# CD47 blockade (ALX301) enhances immunoradiotherapy response in HPV negative head and neck squamous cell carcinoma

Abdula Monther[1,2], Riyam Al-Msari[1,3], Robert Saddawi-Konefka[1,4,5], Santiago Fassardi[1], Cynthia Tang[1], Chad Philips[1], Prakriti Sen[1], Pardis Mohammadzadeh[1], Kelsey Decker[1], Sayuri Miyauchi[1], Souvick Roy[1], Riley Jones[1], Xingyu Wu[1], Silvio Gutkind[1,5,6], Andrew Sharabi[1,5,7], Joseph Califano[1,4,5]*

1 Moores Cancer Center, UC San Diego, La Jolla, California United States of America, 2 University of California-San Diego School of Medicine, La Jolla, California, United States of America, 3 Broad Institute of Harvard and Massachusetts Institute of Technology, Cambridge, Massachusetts, United States of America, 4 Department of Otolaryngology-Head and Neck Surgery, UC San Diego School of Medicine, San Diego, California, United States of America, 5 Gleiberman Head and Neck Cancer Center, UC San Diego, La Jolla, California, United States of America, 6 Department of Pharmacology, UC San Diego, La Jolla, California, United States of America, 7 Department of Radiation Medicine and Applied Sciences, UC San Diego School of Medicine, San Diego, California, United States of America

* jcalifano@health.ucsd.edu

## Abstract

Head and neck squamous cell carcinoma (HNSCC) is a significant cause of morbidity and mortality worldwide, with limited treatment options for patients with locally advanced disease. CD47 immune checkpoint inhibitors have been used to block the CD47/SIRPa interaction that inhibits antigen-presenting cell phagocytosis, thereby enhancing antigen presentation to cytotoxic T-cells, and have shown promise in combination with anti-PD1 immunotherapy in tumors, including recurrent/metastatic HNSCC. We found that CD47 expression is associated with poor prognosis in HNSCC and explored the anti-tumor activity of an anti-CD47 fusion protein in combination with anti-PD1 and lymphatic-sparing radiotherapy in a locally advanced HNSCC model. In the 4MOSC1 syngeneic HPV-negative HNSCC mouse model, ALX301 (an engineered CD47-blocking SIRPα fusion for murine models) induced complete tumor regression when combined with anti–PD-1, and produced a partial tumor response as a monotherapy. An anti-PD1 immune checkpoint inhibitor in a CD47-null tumor background led to complete tumor regression confirming a key role for CD47 in tumor immunity. ALX301 treated mice demonstrated increased MHC-II expression on dendritic cells within the tumor and upregulation of CD86 co-stimulatory molecule on dendritic cells within the tumor, sentinel lymph nodes, and contralateral lymph nodes. Combination ALX301 and anti-PD1 treatment in an anti-PD1 resistant 4MOSC2 model demonstrated significant tumor regression, enhanced survivability, improved response with neoadjuvant radiotherapy, and greater retention of CD8+T-cells within the tumor microenvironment. Notably, T-cell

**Data availability statement:** All TCR sequencing files have been deposited in the NCBI Sequence Read Archive (Accession number PRJNA1372401). All other relevant data are within the paper and its Supporting Information files.

**Funding:** This research was supported by the National Cancer Institute (NCI), www.cancer. gov, project number R01CA281285, awarded to JC and AS. The funders had no role in study design, data collection and analyses, decision to publish, or preparation of the manuscript.

**Competing interests:** The authors have declared that no competing interests exist.

receptor sequencing revealed increased shared clonality between the tumor and sentinel lymph nodes of ALX301 treated mice. These data demonstrate that a combination of CD47 blockade and anti-PD1 therapy enhances tumor antigen presentation and immune cell infiltration, while further improving anti-tumor responses in combination with tumor-targeted radiotherapy. This study provides support for the rational design of combinatorial immunoradiotherapy, using anti-CD47 inhibitors and anti-PD1 therapy, in a clinical trial targeting locally advanced HPV-negative HNSCC.

## Introduction

Head and neck squamous cell carcinomas (HNSCC) arise from the upper aerodigestive tract, including the oral cavity, pharynx, larynx, and paranasal sinuses [1,2]. They represent the seventh most common cancer worldwide, with approximately 890,000 new cases and 450,000 deaths annually [3]. Despite advancement in multimodal therapies including surgery, radiation, chemotherapy, immunotherapy, and rarely targeted therapies, the overall survival rate remains poor for patients with locally advanced diseases [4,5].

Radiotherapy and surgery are primary treatment modalities for patients with locally advanced HNSCC. Therapeutic elective nodal irradiation (ENI) and removal of tumor-draining lymph nodes (neck dissection) are standard of care therapy to reduce regional recurrence in HNSCC [6,7]. Blocking PD-1 interaction with its ligand PD-L1 rescues T cells from exhausted status and revives antitumor immunity [8]. PD-1 inhibitors have shown an overall response rate of 10−20% in recurrent/metastatic HNSCC [9,10]. Recent trials have shown that PD-1 inhibitors can improve event-free survival in locoregionally advanced HNSCC when used neoadjuvantly (before surgery) and continued post-operatively alongside chemotherapy and radiation [11]. However, a clear improvement in overall survival has not been demonstrated. Increasingly, it has become evident that tumor-draining lymph nodes are critical to coordinating the immune response to the primary tumor, and concurrent ENI can abrogate the response to ICI. For example, cytotoxic radio-targeting of the tumor-draining lymph nodes in ENI disrupts dendritic cell-dependent, antigen-specific CD8-driven antitumor immunity, ceasing antitumor response [6,12]. To address this challenge, stereotactic body radiotherapy (SBRT) can deliver ablative high-beam radiation doses while minimizing the volume of irradiated normal tissue and sparing draining lymphatics, facilitating ICI response [13]. Data from clinical trials using tumor-targeted lymphatic-sparing SBRT in HNSCC patients are associated with increased local control and an immunostimulatory tumor microenvironment (TME), specifically when combined with systemic therapy agents such as immune checkpoint inhibitors (ICI) [14,15]. For example, when neoadjuvant SBRT is combined with a prescription immunotherapy anti-PD1 drug such as nivolumab, there is a major pathological response (mPR) rate of 60% in HPV-negative HNSCC as compared to the 7% seen with nivolumab alone [16]. The enhanced pathological response observed may be attributed in part to SBRT induced upregulation of PD-L1 expression in tumors initially PD-L1 negative, thereby rendering

tumors more susceptible to anti-PD1/PD-L1 therapy [17]. Therefore, SBRT immunomodulation may sensitize otherwise unresponsive tumors to immune checkpoint inhibition. However, a critical issue remains: although radiation increases tumor antigenicity and PD-1 blockade reverses T cell exhaustion, tumors can modulate the surrounding microenvironment to escape therapy via upregulation of *other* immunosuppressive markers [18]. CD47 is an immunosuppressive immune checkpoint receptor overexpressed by tumors to evade recognition by the immune system [19]. CD47 interacts with the ligand SIRPα found in myeloid and dendritic cells and conveys a "do not eat me" signal to inhibit tumor phagocytosis, thus blocking ingestion of tumor-specific antigens by professional APCs [19]. Although CD47 is a promising target for ICI, the utility of conventional CD47 blockade is compromised due to on-target, off-tumor phagocytic toxicities caused by extensive expression of CD47 on red blood cells and normal tissue [19]. Some researchers have identified routes around this limitation by using a clinical grade engineered CD47 inhibitor with an inactive Fc region, evorpacept (ALX148), to avoid hematological toxicity [20]. These studies have found that the engineered anti-CD47 inhibitor promotes an antitumor immune response through methods such as increased macrophage phagocytosis, dendritic cell activation, and proinflammatory cytokine production while maintaining a favorable safety profile [20,21]. However, many of these studies do not investigate the lymphatic sparing immunoradiotherapy benefit when combined with anti-CD47 activity. In this study, we evaluate the effects of a mouse-specific engineered CD47-blocking SIRPα fusion protein (ALX301) in combination with anti-PD1 monoclonal antibody and tumor-directed radiation therapy in preclinical HPV negative HNSCC models. We aim to determine whether CD47 inhibition can enhance antigen presentation, T-cell activation, and ultimately improve tumor control, thereby providing rationale for new immunoradiotherapy combinations in HSCC. We hypothesized that ALX301 would improve the tumor response to PD-1 inhibition and radiotherapy by promoting phagocytosis of tumor antigens and enhancing their presentation to T cells. This, in turn, would expand the tumor-specific T-cell repertoire and strengthen cytotoxic T-cell–mediated tumor cell killing. We then placed these strategies in the context of a model for combinatorial immunoradiotherapy designed to spare injury to tumor draining lymphatics and thereby enhancing antitumor response.

## Results

### CD47 confers poor prognosis and immune evasion

To explore the clinical relevance of CD47 expression in HNSCC, we used patient data from the Cancer Genome Atlas (TCGA) dataset. Kaplan-Meier survival analysis showed that patients with higher CD47 expression had significantly lower disease-free survival than did those with lower expression, suggesting that CD47 may serve as a marker of poor prognosis (Fig 1A, p(HR) = 0.026). These findings are consistent with other studies finding that lower expression of CD47 led to a significant increase in survivability [22]. To evaluate the impact of CD47 expression in preclinical HNSCC models, we employed previously characterized 4-NQO carcinogen-induced, 4MOSC1 murine oral squamous cell carcinoma cancer cell lines that can be employed in syngeneic, orthotopic models. This model displays a human tobacco-signature mutanome in addition to an immune infiltrate and ICI response pattern similar to that observed clinically in HPV-negative HNSCC, with partial response to anti-PD1 therapy [23]. To define the role of CD47 in this model, we created a CD47 knockout from our 4MOSC1 cell line to investigate the effect on tumor growth and the TME, with the protocol and primer sequences outlined in Methods. We used a selective CRISPR antigen removal lentiviral vector system to ensure the elimination of Cas9 from the genome later, preventing immunogenicity [24]. Cas9 immunogenicity had been verified in Cas9-expressing 4MOSC1 buccal tumor-bearing mice as seen with the significant tumor regression compared to the control (Fig 1B, p < 0.0001). Knockout of CD47 and Cas9 was verified (Fig 1C). We observed that knocking CD47 out of 4MOSC1 sensitizes the cell line, resulting in complete tumor regression, defined as complete or almost complete tumor eradication (Fig 1D, parent vs. CD47KO p < 0.0001). Using the CD47 KO Cas9 KO 4MOSC1 cell line, we further observed complete tumor regression in the knockout groups, whereas the parent 4MOSC1 cell line tumors continued to grow (Fig 1E, parent vs. Cas9KO CD47KO p < 0.0001).

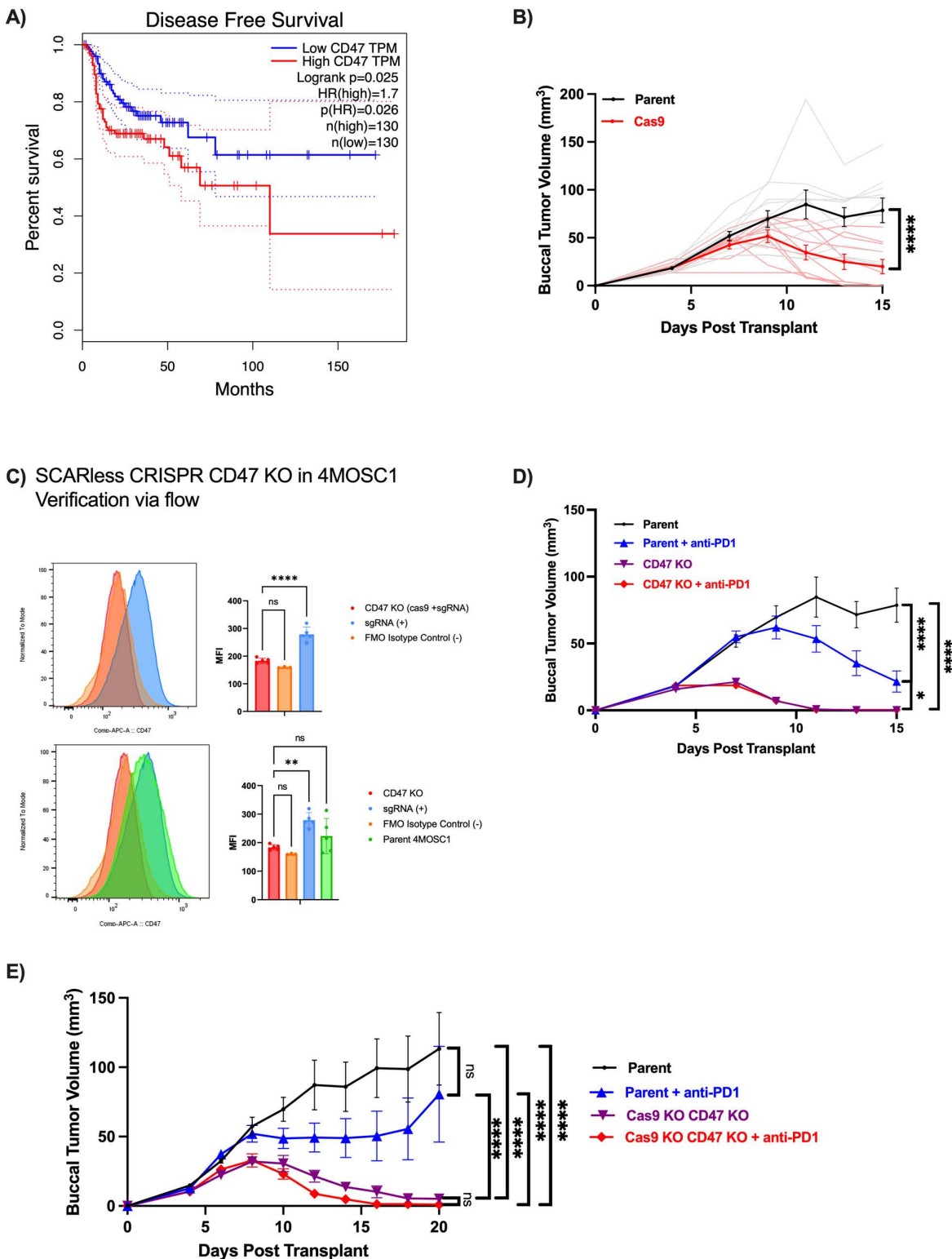

**Fig 1. CD47 Expression Confers Poor Prognosis in HNSCC and Knockout Enhances Tumor Immunogenicity. (A)** Kaplan Meier survival analysis of TCGA HNSC data set based on CD47 expression [25]. A quartile cut-off was used and survival analysis was performed using log-rank tests. **(B)** Representative tumor growth kinetics of mice with tumors of the parent 4MOSC1 cell line vs. tumors with Cas9 expressing 4MOSC1 cell line (*n* = 10 mice per

group; $p < 0.0001$). The $p$ value was calculated using a two-way repeated measures ANOVA test with the source of variation being time x column factor. Data are presented as mean values ± SEM. In the Cas9 group, 5/10 mice had complete tumor eradication and 3/10 had tumor reduction. **(C)** Flow cytometry results demonstrating knockout of CD47 and Cas9 from the 4MOSC1 parent cell line ($n = 5$ for CD47 KO, $n = 3$ for FMO Isotype Control (-), $n = 5$ for sgRNA (+), $n = 5$ for Parent 4MOSC1; **$p = 0.0047$, ****$p < 0.0001$, p = 0.7813 for CD47 KO vs. FMO Isotype Control (top), p = 0.8114 for CD47 KO vs. FMO Isotype Control (bottom), p = 0.3401 for CD47 KO vs. Parent 4MOSC1 (bottom)). Data are presented as mean ± SD; $p$ values were calculated using ordinary one-way ANOVA with Tukey's post-hoc. FMO isotype controls are used to determine the cut-off point between background fluorescence and positive populations in multi-color immunofluorescent experiments. 4MOSC1 cells were first transduced with pSCAR_Cas9 GFP-expressing vector, infected with pSCAR_sgRNA targeting CD47, then cultured for 10 days to allow Cas9 genome editing and removal of CD47, and later the IDLV-Cre vector expressing Cre recombinase was transduced to excise the loxP site containing Cas9. **(D)** Representative tumor growth kinetics of mice with 4MOSC1 parent tumors vs. 4MOSC1 parent tumors treated with anti-PD1 vs. 4MOSC1 CD47 knockout tumors vs. 4MOSC1 CD47 knockout tumors treated with anti-PD1 ($n = 10$ mice per group; $p = 0.0453$ for Parent + anti-PD1 vs. CD47 KO, $p > 0.99$ for CD47 KO vs. CD47KO + anti-PD1, $p = 0.0515$ for Parent + anti-PD1 vs. CD47KO + anti-PD1, ****$p < 0.0001$). Data are presented as mean ± SEM; $p$ values were calculated using ordinary two-way ANOVA with Tukey's post-hoc. In the 4MOSC1 Parent group, 2/10 mice had a partial reduction in tumor size, while 8/10 had continuous tumor growth. In the 4MOSC1 Parent + anti-PD1 group, 3/10 mice had complete tumor eradication, 6/10 mice had tumor reduction, and 1/10 mice had continuous tumor growth. In the 4MOSC1 CD47KO group, 10/10 mice had complete tumor eradication. In the 4MOSC1 CD47KO + anti-PD1 group, 9/10 mice had complete tumor eradication, and 1/10 mice had tumor reduction. **(E)** Representative tumor growth kinetics of mice with 4MOSC1 parent tumors vs. 4MOSC1 parent tumors treated with anti-PD1 vs. 4MOSC1 CD47 Cas9 knockout tumors vs. 4MOSC1 CD47 Cas9 knockout tumors treated with anti-PD1 are shown ($n = 10$ mice per group; $p = 0.1669$ for Parent vs. Parent + anti-PD1, $p = 0.9935$ for 4MOSC1 Cas9KO CD47KO vs. 4MOSC1 Cas9KO CD47KO + anti-PD1, ****$p < 0.0001$). Data are presented as mean ± SEM; $p$ values were calculated using ordinary two-way ANOVA with Tukey's post-hoc. In the 4MOSC1 Parent group, 1/10 mice had spontaneous tumor eradication, and 9/10 mice had continuous tumor growth. In the 4MOSC1 Parent + anti-PD1 group, 3/10 mice had complete tumor eradication, 2/10 had tumor reduction, and 5/10 mice had continuous tumor growth. In the 4MOSC1 CD47KO Cas9KO group, 6/10 mice had complete tumor eradication, and 4/10 mice had tumor reduction. In the 4MOSC1 CD47KO Cas9Ko group, 9/10 mice had complete tumor eradication and 1/10 mice had tumor reduction.

## ALX301 therapy enhances tumor control and synergizes with radiation and immune checkpoint blockade

Based on the results obtained from CD47 4MOSC1 knockout models, we chose to observe how these findings would translate through CD47 inhibition obtained through pharmacologic CD47 blockade using ALX301. ALX301 is an engineered CD47-blocking SIRPα fusion protein with an N297A mutation which minimizes interaction with the myeloid cell Fc receptor, decreasing off-tumor toxicity [19]. To explore the potential role of pharmacologic anti-CD47 treatment, we defined response to ALX301 in models employing the anti-PD1 partially responsive 4MOSC1 model as well as an anti-PD1 resistant 4MOSC2 mode [23]. Mice transplanted with the 4MOSC1 or 4MOSC2 cell line were treated with either ALX301 (30 mg/kg every 4 days starting on day 6), anti-PD1 (10 mg/kg on days 6 and 8), or both and monitored for tumor growth/regression. Combination treatment of ALX301 with anti-PD1 led to a complete tumor regression in the 4MOSC1 model (Fig 2A), but only a partial tumor regression in the 4MOSC2 cell line, with partial tumor regression defined as tumor reduction but not complete tumor eradication (Fig 2B). To overcome the partial tumor regression observed in the 4MOSC2 cell line, a triple regimen of ALX301, anti-PD1, and tumor-directed radiation therapy (tdRT) was designed (Fig 2C). The addition of tdRT therapy was based on the immunomodulatory effects observed after SBRT was implemented in both preclinical and clinical settings. A one-time dose of 4Gy was chosen for tdRT based on results that provided evidence that this dose of tdRT was not cytotoxic but still enhanced the percentage of CD8 + T cells and dendritic cell trafficking to the 4MOSC tumor [26]. With SBRT, there was an enrichment of tumor-specific T cells and CD8 + cytotoxic T cells, indicating an enhancement of an antitumor response [15]. Furthermore, we chose to deliver the tdRT before ICI administration due to evidence that delivering irradiation treatment directly to the gross tumor and sparing the cervical lymphatics can improve the systemic response that anti-PD1 and other immunotherapies provide [12,16]. Fig 2C shows that adding tumor-directed radiation therapy (tdRT) to the treatment regimen led to further tumor regression in the 4MOSC2 model. Treatment with anti-PD1, ALX301, or a combination, along with tdRT increased long-term survivability ($n = 10$ for each arm, control vs. anti-PD1 + tdRT p = 0.0505, control vs. anti-PD1 + ALX301 p = 0.0114*, control vs. anti-PD1 + ALX301 + tdRT p = 0.016*; S1 Fig 1C). Notably, only treatment groups that included ALX301 showed significantly prolonged survival, underlining the contribution of CD47 blockade to improved outcomes. A comparison between ALX301 and tdRT therapies in the 4MOSC2 model as monotherapies and combinatory therapies was performed showing partial reduction with tdRT

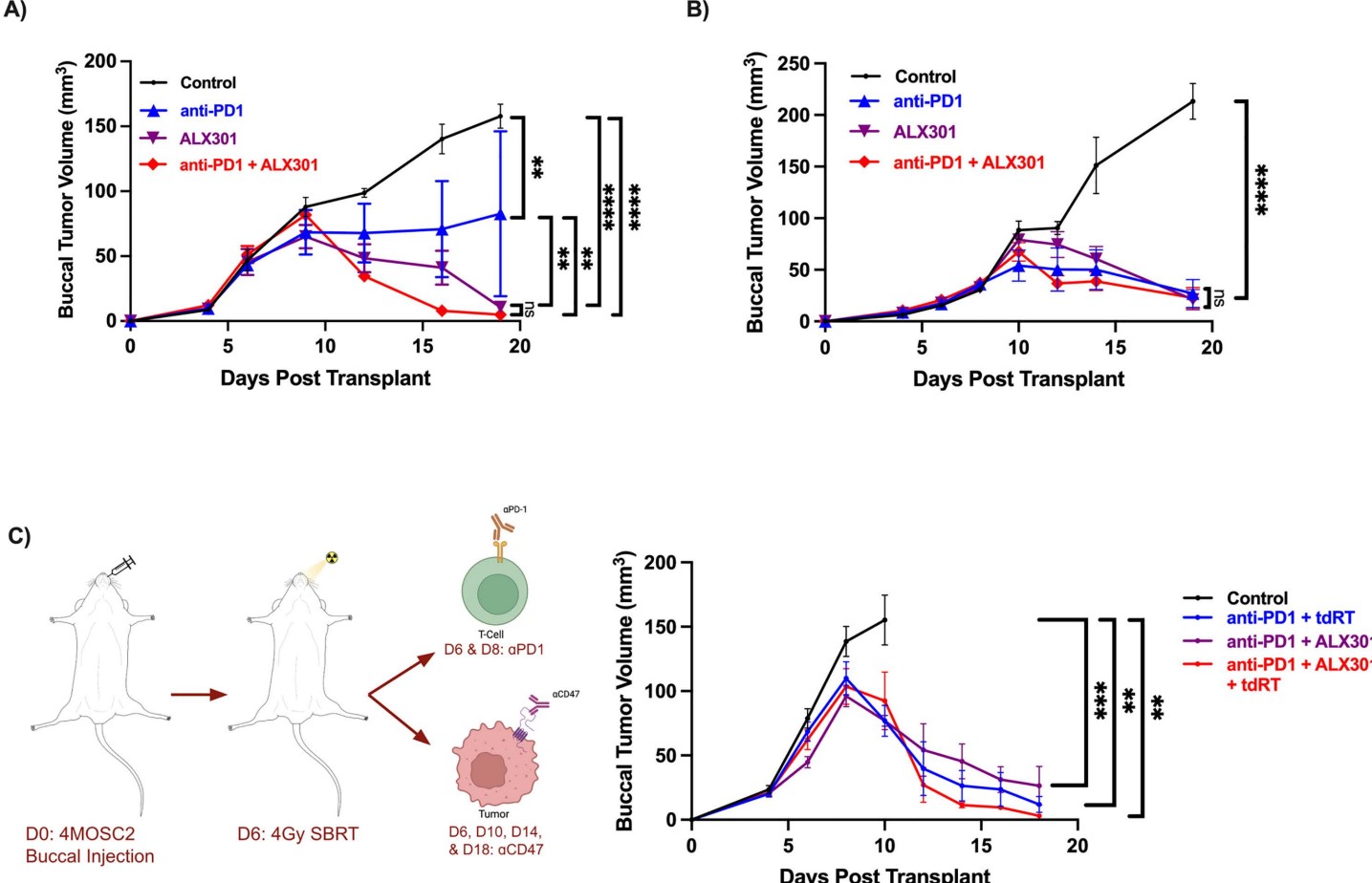

**Fig 2. Synergistic Antitumor Effects of ALX301, anti-PD1, and Tumor-Directed Radiation Therapy. (A)** Representative tumor growth kinetics of mice with 4MOSC1 tumors (control) vs. 4MOSC1 tumors treated with anti-PD1 vs. 4MOSC1 tumors treated with ALX301 vs. 4MOSC1 treated with both ALX301 and anti-PD1 are shown ($n = 5$ mice per group up to day 16, $n = 4$ mice for anti-PD1 on day 19 due to one mouse succumbing to disease; **$p = 0.0032$ for Control vs. anti-PD1, **$p = 0.0053$ for anti-PD1 vs. ALX301, **$p = 0.0022$ for anti-PD1 vs. anti-PD1+ALX301, $p = 0.9919$ for ALX301 vs. anti-PD1+ALX301, ****$p < 0.0001$). Data are presented as mean±SEM; $p$ values were calculated using ordinary two-way ANOVA with Tukey's post-hoc. In the control group, 5/5 mice had continuous tumor growth. In the anti-PD1 treated group, 1/5 mice had complete tumor eradication, 2/5 had tumor reduction, 1/5 succumbed to disease, and 1/5 had continuous tumor growth. In the ALX301 group, 1/5 mice had complete tumor eradication and 4/5 mice had tumor reduction. In the ALX301+anti-PD1 group, 3/5 mice had complete tumor eradication and 2/5 mice had tumor reduction. **(B)** Representative tumor growth kinetics of mice with 4MOSC2 tumors (control) vs. 4MOSC2 tumors treated with anti-PD1 vs. 4MOSC2 tumors treated with ALX301 vs. 4MOSC2 treated with both ALX301 and anti-PD1 are shown ($n = 5$ for control up to day 14, $n = 4$ for control on day 19. $n = 5$ for anti-PD1 up to day 14, $n = 4$ for anti-PD1 on day 19. $n = 6$ for ALX301 up to day 14, $n = 5$ for ALX301 on day 19. $n = 6$ mice for anti-PD1+ALX301. Mice were omitted due to succumbing to disease; ****$p < 0.0001$ for Control vs. all other groups, $p > 0.97$ when comparing experimental groups to each other). Data are presented as mean±SEM; $p$ values were calculated using ordinary two-way ANOVA with Tukey's post-hoc. In the control group, 4/5 mice had continuous tumor growth, and 1/5 mice had succumbed to disease. In the anti-PD1 group, 1/5 mice had complete tumor eradication, 1/5 had tumor reduction, 2/5 had continuous tumor growth, and 1/5 had succumbed to disease. In the ALX301 group, 1/5 mice had continuous tumor growth, 4/5 had tumor reduction, and 1/5 had succumbed to disease. In the anti-PD1+ALX301 group, 5/6 mice had tumor reduction, and 1/5 had continuous tumor growth. **(C)** Left: Graphic representation of the experimental design for the triple regimen of anti-PD1, ALX301, and tdRT. The 4MOSC2 cell line will be transplanted into the buccal mucosa on day 0 for all mice. Mice who receive tdRT will receive 4Gy of tdRT on day 6. Mice who receive anti-PD1 treatment will receive it on day 6 and day 8, and mice who receive ALX301 treatment will receive it on day 6, day 10, day 14, and day 18. Right: Representative tumor growth kinetics of the experimental design shown to the left with 4MOSC2 (control) vs. 4MOSC2 tumors treated with anti-PD1 and tdRT vs. 4MOSC2 tumors treated with anti-PD1 and ALX301 vs. 4MOSC2 treated with anti-PD1, ALX301, and tdRT (For Control: $n = 10$ up to day 6, $n = 9$ on day 8, and $n = 3$ on day 10. For anti-PD1+tdRT: $n = 10$ up to day 8, $n = 6$ on day 10, $n = 3$ from days 12 to 18. For anti-PD1+ALX301: $n = 10$ up to day 8, $n = 8$ on day 10, $n = 3$ from days 12 to 18. For anti-PD1+ALX301+tdRT: $n = 10$ up to day 8, $n = 8$ on day 10, $n = 2$ from days 12 to 18. Mice were omitted due to succumbing to disease; **$p = 0.0091$ for Control vs. anti-PD1+tdRT, ***$p = 0.0007$ for Control vs. anti-PD1+ALX301, **$p = 0.0027$ for Control vs. anti-PD1+ALX301+tdRT, $p = 0.8458$ for anti-PD1+tdRT vs. anti-PD1+ALX301, $p = 0.9778$ for anti-PD1+tdRT vs. anti-PD1+ALX301+tdRT, $p = 0.9778$ for anti-PD1+ALX301 vs.

anti-PD1+ALX301+tdRT). Data are presented as mean±SEM; *p* values were calculated using ordinary two-way ANOVA with Tukey's post-hoc. In the control group, 10/10 mice had continuous tumor growth before all succumbed to disease. In the anti-PD1+tdRT group, 3/10 mice had tumor reduction, and 7/10 succumbed to disease. In the anti-PD1+ALX301 group, 3/10 mice had tumor reduction, and 7/10 mice succumbed to disease. In the anti-PD1+ALX301+tdRT group, 2/10 mice had tumor reduction, and 8/10 mice succumbed to disease.

as a monotherapy and significant reduction with ALX301 as dual therapy (S1 Fig 1G). Based on these findings, there is evidence that an anti-CD47 ICI in combination with either an anti-PD1 ICI or 4 Gy tdRT can overcome an immunosuppressive TME.

## ALX301 therapy enhances dendritic cell mediated T-cell responses

To better understand the mechanism by which anti-CD47 ICI can provide antitumor immunity, we characterized tumor, sentinel lymph nodes (SLN, the first lymph node that receives lymph flow from the primary tumor in a regional lymphatic basin), and contralateral lymph nodes (CLN, a lymph node from the regional lymphatic basin on the contralateral side) on a cellular level. We first transplanted mice with 4MOSC1 tumors and treated one group with the vehicle and the other with 3 doses of ALX301, where dosages were administered every 4 days starting on day 6 at a concentration of 30 mg/kg. On day 15 post-transplant, the SLN and CLN were mapped using lymphazurin to visually confirm the tumor draining sentinel lymph node (SLN) and then harvested along with the tumor (Fig 3A). ALX301 inhibited tumor growth (Fig 3B, *p*=0.0003), and we noted upregulation in MHC II in CD11c+MHC II+ dendritic cells (DC) in the tumor (Fig 3C). This suggests that ALX301 enhanced immune activation through DCs being able to more effectively present antigens to T cells, leading to a stronger T cell-mediated tumor regression. We also noted an increase in CD86 expression on activated DCs in the tumor, SLN, and CLN (Fig 3D), suggesting a co-stimulatory signal on activated CD4+T cells that works synergistically with MHC II to activate and prime T cells for an anti-tumor response. Taken together, these results demonstrate that ALX301 treatment as a standalone therapy elicits an enhanced immune response through upregulation of MHC II and CD86 on DCs allowing for greater antigen presentation, T-cell activation, and T-cell expansion. While increased MHC II and CD86 expression is consistent with a more activated dendritic cell phenotype, our study does not directly demonstrate cross-priming of tumor-specific cytotoxic T cells by DCs. Of note, the effect of ALX301 on other immune parameters, including the count of CD8 T-cells in the tumor, CD86 MFI in F4/80+macrophages, and percent CCR7+activated DCs were not statistically demonstrated.

To explore how ALX301 can be combined with other therapeutic modalities, we employed our ICI resistant 4MOSC2 model and treated it with ALX301, anti-PD1, and tdRT in a fashion similar to that depicted in Fig 2C. (Tumor Kinematics S1 Fig 1E and S1 Fig 1F). Tumors of the mice in all treatment groups demonstrated upregulation of CD69 in CD8+T cells (Fig 4), consistent with the effects of anti-PD1 in upregulating T cell activation within the TME as well as in increasing T cell retention in the tumor for a sustained local immune response [27]. Additionally, we did not demonstrate changes in CD69 MFI in natural killer (NK) cells, macrophage recruitment (F4/80+Mac% in CD45+), and CD86 MFI in F4/80+macrophages, consistent with a primary effect of treatment on end effector CD8+T cells.

## ALX301 therapy promotes expansion and migration of expanded T-cell clonotypes

To specifically assess how treatment with ALX301, anti-PD1, and tdRT affects the T-cell response, we performed TCR sequencing on the tumors and SLNs of mice treated above (Figs 3 and 4). To match the results of the results of the TCR seq to the results of the flow cytometry, one tumor and SLN were extracted from each group. TCRa and TCRb clonality analysis demonstrates that in both 4MOSC1 and 4MOSC2 models, there is enhancement in clonal expansion in ALX301 treated mice, with the one exception of 4MOSC2 anti-PD1+tdRT tumors (which may be due to a large therapeutic response to the combination therapy of anti-PD1 and tdRT). We also defined overlap of TCRa and TCRb clonotypes between the tumors and SLN in the 4MOSC1 control vs. ALX301 experiment (Figs 5B and 5C) showing an increase in

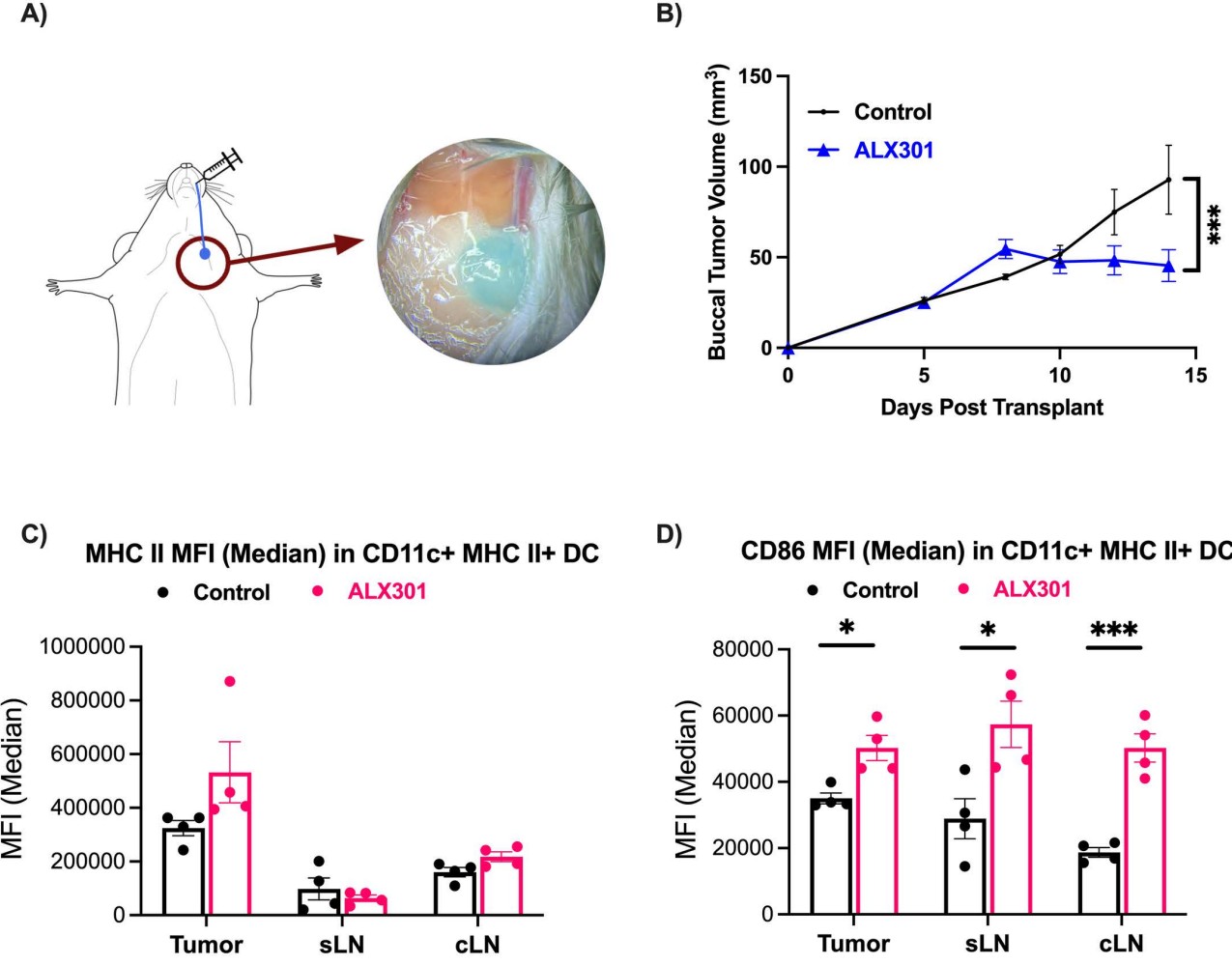

**Fig 3. ALX301 Therapy Reduces Tumor Burden and Enhances Immune Activation.** Mice were injected with 4MOSC1 cells ($1\times10^6$ cells) in the buccal mucosa, and on days 6, 10, and 14 were given 30 mg/kg of ALX301 before being assessed on Day 15. **(A)** Process of lymphatic mapping of the cervical draining lymph node in the 4MOSC1 mouse model. Lymphazurin is injected into the buccal mucosa and the draining lymph node is marked blue. **(B)** Representative tumor growth kinetics of mice with 4MOSC1 tumors (control) vs. 4MOSC1 tumors treated with ALX301 ($n=5$ mice per group; ***$p=0.0003$). Data are presented as mean±SEM; $p$ values were calculated using a two way repeated measures ANOVA test with the source of variation being time x column factor. In the control group, 5/5 mice had continuous tumor growth. In the ALX301 group, 5/5 mice had tumor reduction. **(C)** In the tumors of the ALX301 treated group, there was an upregulation in MHC II in CD11c+MHC II+ dendritic cells ($n=4$ per group; $p=0.1268$ for tumors, $p=0.4576$ for SLN, $p=0.0659$ for CLN). Data are presented as mean±SEM; $p$ values were calculated using a two-tailed unpaired t-test. **(D)** In the tumors, SLNs, and CLNs of the ALX301 treated group, there was an upregulation in CD86 in CD11c+MHC II+ dendritic cells ($n=4$ per group; *$p=0.0104$ for tumor, *$p=0.0213$ for SLN, ***$p=0.0004$ for CLN). Data are presented as mean±SEM; $p$ values were calculated using a two-tailed unpaired t-test.

common TCR clones in ALX301 treated tumors. Notably, in 4MOSC2 models, we saw greater shared clonality between the tumors and SLNs in mice treated with ALX301, and a greater Morisita overlap index between 4MOSC2 tumor and SLN in the anti-PD1+ALX301 treated arms (Figs 5D, 5E, and 6). The increase in shared clonality between the tumors and SLNs indicated that the T cells that were activated and expanded within SLN in response to ALX301 had migrated to the TME, potentially further amplifying the immune response. This also reinforces the key role of antigen exposed, migratory DCs to the SLN and draining lymph nodes in antigen specific activation of tumor-infiltrating lymphocytes (TILs) and response to tumor-associated antigens. Ultimately, these results suggest that CD47 blockade plays a critical role in the

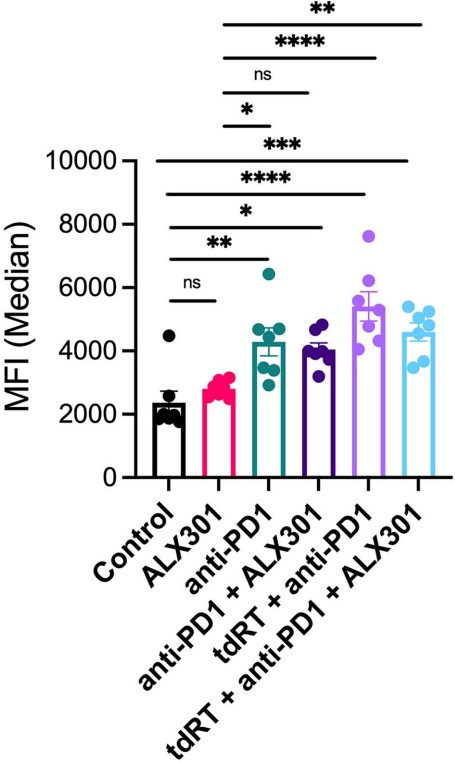

**Fig 4. Synergistic Therapy Using ALX301, anti-PD1, and Tumor-Directed Radiation Therapy Upregulates CD69 in CD8+T Cells.** Mice were injected with 4MOSC2 cells (0.5x10⁶ cells) in the buccal mucosa before being assessed on Day 9. Mice in the anti-PD1 and all combination treatment groups observed an upregulation of CD69 in CD8+T cells in the tumor ($n=7$ tumors per group; p=0.9381 for Control vs. ALX301, **$p=0.0034$ for Control vs. anti-PD1, *$p=0.0135$ for Control vs. anti-PD1+ALX301, ****$p<0.0001$ for Control vs. tdRT+anti-PD1, ***$p=0.0005$ for Control vs. tdRT+anti-PD1+ALX301, *$p=0.0385$ for ALX301 vs. anti-PD1, p=0.1206 for ALX301 vs. anti-PD1+ALX301, ****$p<0.0001$ for ALX301 vs. tdRT+anti-PD1, **$p=0.0072$ for ALX301 vs. tdRT+anti-PD1+ALX301, p=0.9956 for anti-PD1 vs. anti-PD1+ALX301, p=0.2008 for anti-PD1 vs. tdRT+anti-PD1, p=0.9860 for anti-PD1 vs. tdRT+anti-PD1+ALX301, p=0.0706 for anti-PD1+ALX301 vs. tdRT+anti-PD1, p=0.8530 for anti-PD1+ALX301 vs. tdRT+anti-PD1+ALX301, p=0.5407 for tdRT+anti-PD1 vs. tdRT+anti-PD1+ALX301). Data are presented as mean±SEM; *p* values were calculated using ordinary one-way ANOVA with Tukey's post-hoc.

expansion and migration of specific T-cell clonotypes in response to tumor-associated antigens, and that CD47 blockade can specifically enhance anti-tumor specific T cell repertoire.

## Discussion

The rise of ICI therapy such as immunotherapy targeting PD1/PD-L1 has shown promise as an effective treatment with low toxicity [28]. Despite the overall increase in survivability in patients who receive PD1 ICI treatments, recent studies demonstrate that most patients with HNSCC may not respond to PD1 ICI therapy or will develop resistance to the immunotherapy after some time [29]. In the Javelin and Head Neck 100 trials, it was demonstrated that avelumab failed to improve the progression-free survival and overall survival of HNSCC compared to other types of cancers, such as non-small cell lung cancer, which saw improved outcomes [30]. These data suggest that we need additional therapies that can be as effective as, or more effective than, current options – either used alone or in combination with ICIs [30]. Clinical trials also demonstrate the importance of treatment sequencing. For example, delivering anti–PD1 before chemoradiotherapy (CRT) can abrogate the immune response (due to radiosensitive CD8 T cells), whereas delivering anti–PD1 after CRT

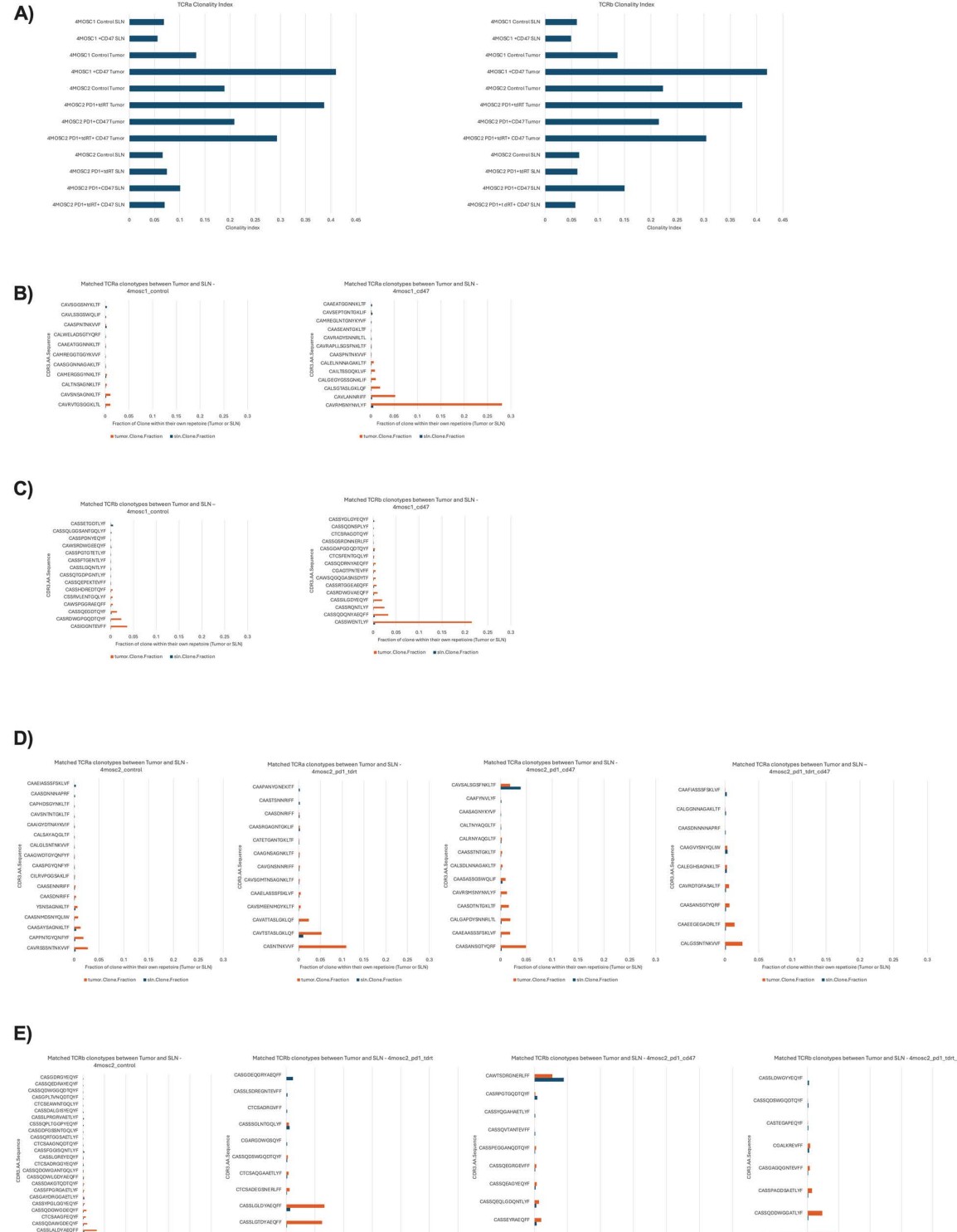

**Fig 5. ALX301 Therapy Enhances Clonal Expansion in 4MOSC1 Tumors.** TCR sequencing of 1 tumor and 1 SLN from each treatment group in Fig 3 (4MOSC1 control vs. ALX301) and Fig 4 (4MOSC2 control vs. anti-PD1+ALX301 vs. anti-PD1+tdRT vs. anti-PD1+ALX301+tdRT). Labels with "PD1" refer to anti-PD1 therapy, "tdRT" refer to tumor-directed radiation therapy, and "CD47" refer to ALX301 therapy. TCR sequencing demonstrates that

ALX301 treatment leads to upregulation of clonal expansion within the tumor and shared clonality between the tumors and SLNs. **(A)** TCRa and TCRb Clonality Index of the tumors and SLNs of the different models and treatment groups. **(B)** Matched TCRa clonotype between tumors and SLNs of the 4MOSC1 model. **(C)** Matched TCRb clonotype between tumors and SLNs of the 4MOSC1 model. **(D)** Matched TCRa clonotype between tumors and SLNs of the 4MOSC2 model. **(E)** Matched TCRb clonotype between tumors and SLNs of the 4MOSC2 model.

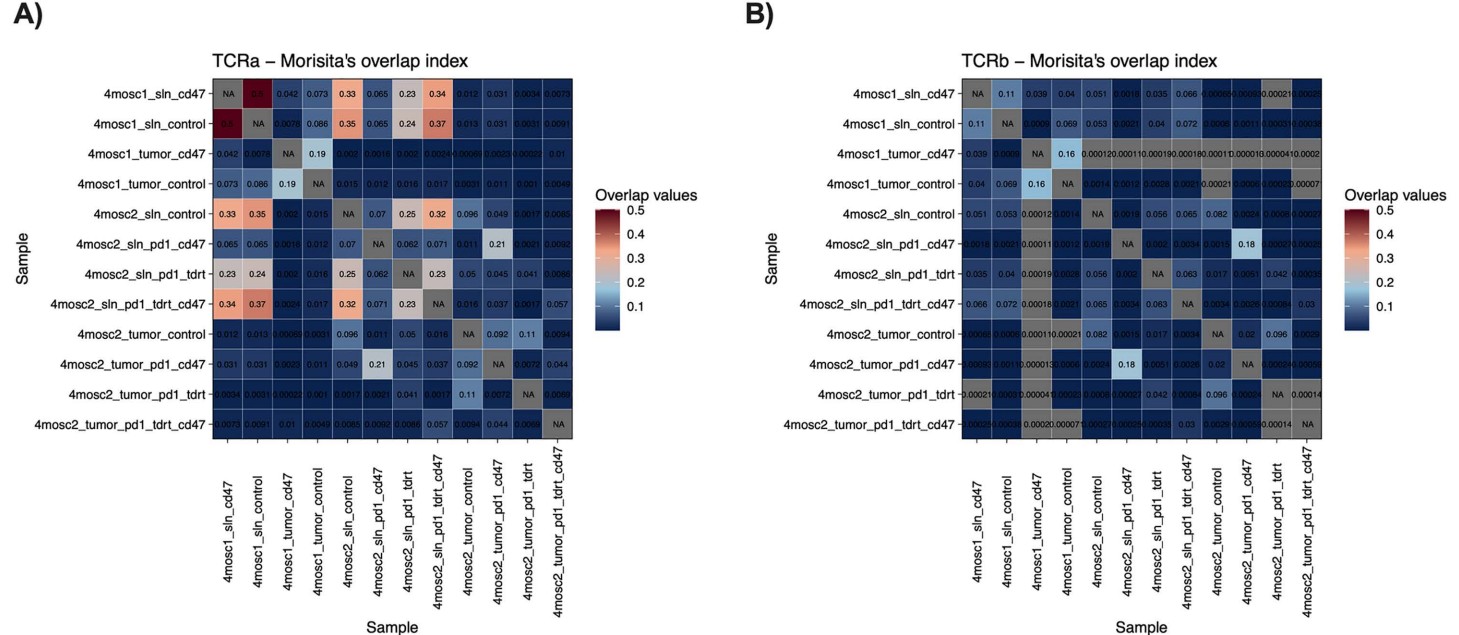

**Fig 6. ALX301 With anti-PD1 Therapy Increases Overlap of T-Cell Receptors Between Tumor and Sentinel Lymph Node.** Morisita's Overlap Index of the TCR-sequencing data observed in Fig 5 demonstrating increased shared clonality between tumors and SLNs of ALX301 treated groups. Labels with "PD1" refer to anti-PD1 therapy, "tdRT" refer to tumor-directed radiation therapy, and "CD47" refer to ALX301 therapy. **(A)** TCRa clonality for both 4MOSC1 and 4MOSC2 models. **(B)** TCRb clonality for both 4MOSC1 and 4MOSC2 models.

yields better tumor control [30]. Not only was the sequencing of the combined therapies found to be important, but also whether the tumor-draining lymph nodes were included in the radiation fields. In clinical treatments of HNSCC, current evidence shows that the antitumor immune response derived from PD-1/PD-L1 inhibitors results from the CD8+T cells within the tumor-draining lymph nodes [31]. When these lymph nodes are included in the radiation field, it leads to a depletion of antitumor CD8+T cells, therefore inhibiting the host's antitumor immunity [30]. Given these considerations, we designed a new combinatorial therapy approach in our study. As CD47, another immune checkpoint located upstream of PD-1 in the adaptive tumor immune response pathway, is highly expressed in many tumors and is targetable, this presents a good candidate for combination treatment along with PD1 ICIs. Moreover, these therapies paired with SBRT could target the tumor without exposing the lymph nodes to the elective radiation volume, further optimizing treatment.

CD47 is an immune checkpoint that delivers a 'Don't Eat Me' signal to phagocytes via SIRPα binding, and it is overexpressed in many tumors to enable immune evasion [32]. Based on these findings, a clinical grade CD47-blocking myeloid checkpoint inhibitor, evorpacept (ALX148), was developed to promote phagocytosis of tumor cells. It was found to have a favorable safety profile in clinical studies [20]. Additionally, in preclinical models, evorpacept combined with anti-PD1 led to favorable outcomes such as increased tumor growth inhibition and extended survivability [21]. While phase 2 clinical trials investigating evorpacept, pembrolizumab, and chemotherapy combination therapy for locally advanced HNSCC failed to meet primary endpoints, these findings suggest that additional or alternative therapeutic partners may be necessary for

the synergistic potential of CD47 and PD1 inhibition (NCT05787639 and NCT04675333) [33]. Based on these results, we chose to explore a similar anti-CD47 ICI (ALX301) developed for murine models in combination with other therapeutics while avoiding lymph node ablation.

In our study, after confirming the prognostic association of CD47 expression in HNSCC, we first confirmed the effectiveness of CD47 as a viable target for immunotherapy. After knocking CD47 out of the 4MOSC1 genome, we demonstrated that the CD47 knockout tumor cells were significantly more susceptible to anti-tumor immune response than the parent tumor cells even when the parent tumor was treated with an anti-PD1 ICI (Fig 1D, parent + anti-PD1 vs. CD47KO p = 0.0453). After establishing that CD47 is a viable target for HNSCC therapeutics, we used an engineered CD47-blocking SIRPα fusion protein (ALX301) with an N297A mutation to treat 4MOSC1/4MOSC2 HNSCC in a murine model and demonstrated its effectiveness as a standalone therapeutic as well as in combination with an anti-PD1 ICI. In our 4MOSC1 tumor model, we found that treating mice with either anti-PD1 or ALX301 monotherapies had similar responses and provided better results than the control group. However, with a dual therapy of anti-PD1 and ALX301, we were able to achieve complete regression of 4MOSC1 tumors. In our anti-PD1 resistant 4MOSC2 model, using dual therapy of anti-PD1 and ALX301 showed significant response compared to the control (Fig 2C, p = 0.0007), and groups with ALX301 as part of their treatment demonstrated enhanced survival. Interrogation of the immune microenvironment showed that in ALX301 treated 4MOSC1 tumors there was an increase in activated dendritic cells in tumors and draining sentinel nodes, accompanied by an increase in activated T cells in the TME. When we combined ALX301 with anti-PD1 and tdRT as a triple regimen against our aggressive 4MOSC2 model, we also saw increased activation and retention of CD8 + T cells. After performing TCR sequencing on these models, we observed that ALX301 treatment increased overall T-cell clonal expansion and the fraction of dominant T-cell clones, as well as increased shared TCR clonality between tumor and SLN. This finding suggests a more robust, shared antigen-specific T-cell response developing in the lymph node and homing to the tumor. It is important to note that these observations, while suggestive of our hypothesis that ALX301 increases trafficking and migration of APCs, are based on limited sampling.

These findings support the rationale for CD47 inhibition in combination with current immunomodulatory treatments such as anti-PD1 ICIs and SBRT therapies. HPV-negative HNSCC is associated with a poor prognosis, so it is crucial to define novel, effective therapeutic strategies that can enhance response [2]. By demonstrating that anti-PD1 and radiotherapy can be further optimized in combination with anti-CD47, we have provided a basis for rational combination of these therapies in future clinical trials to enhance tumor response.

Although we have discovered promising results in our treatment of the syngeneic HPV-negative HNSCC, animal models are always limited: they do not capture the full complexity of the disease in a clinical setting. Moreover, due to the short duration of treatment of the 4MSOC tumors, we were unable to evaluate the toxicity of the regimen. Past studies have demonstrated that CD47 blockade expresses minimal drug toxicity due to the lack of pro-phagocytic signals on healthy cells [34]. Throughout our study, no toxicity was observed in mice who received ALX301 therapy. It is important to note that while mice and humans share many aspects of cellular identity, key differences exist that impact the immune response to cancer and CD47 blockade [35]. Such differences include expression and binding of selectins, cytokine production, and Fc receptor expression [35]. Therefore, future studies that investigate the long-term toxicity profile of anti-CD47 are warranted to ensure safety during prolonged treatment in a clinical setting. Additionally, while the 4MOSC model displays the mutational profile, immune landscape, and ICI responsiveness of HPV-negative HNSCC, our findings may not extend to all HPV-negative tumors. Another limitation is the T-cell receptor sequencing analysis, which was performed on a single sample per group. These results provide a snapshot of clonality to provide insight into the flow cytometry results of the tumors and lymph nodes, but not statistical results. Future studies with larger sample sizes are warranted to confirm the observed clonal expansion patterns with statistical comparisons.

In HNSCC, the mechanism of CD47 inhibition, and the efficacy with which it impacts tumor progression, is still largely unknown. Previous studies demonstrate that CD47 is overexpressed in HNSCC and that its inhibition as a standalone therapy has led to the stimulation of effector T cells and a decrease in suppressive immune cells [34]. As we have

demonstrated in our own study, the benefits of CD47 inhibition as a combination therapy reveal its promise as part of future HNSCC treatment strategies, pending clinical validation. CD47 inhibition is not limited to combination with anti-PD1 and tdRT: it has shown great results synergistically with other therapies as well. One of the most prominent cancer therapeutics, tumor targeted antibodies, has been shown to be enhanced when combined with anti-CD47, notably because of the enhanced phagocytosis from anti-CD47 in combination with antibody-dependent activation of macrophages through the Fc-gamma receptor [36]. These findings, along with our own, illustrate the multiple advantages of introducing a CD47 inhibition into the range of current practices as a method of enhancing antitumor activity.

As clinical studies investigating anti-CD47 therapies progress, leveraging syngeneic HNSCC models to derive rational sequencing and design of combination therapies can help accelerate and optimize clinical trial design. In addition, mechanistic understanding of the ways in which CD47 inhibition interacts with current therapies, including those that affect draining lymphatic basins, can help to synergize CD47 inhibition with current standard of care therapy [12]. Our results indicate that CD47 blockade, alone or in combination with anti-PD1 and radiotherapy, enhances dendritic cell activation and T-cell expansion in murine HPV-negative HNSCC models. The increased clonality and overlap between tumors and draining lymph nodes suggest that CD47 inhibition may influence antigen-specific immune trafficking between the tumor and lymph node, providing rational to lymphatic preservation. While these effects were observed in mice, they point to mechanisms that could inform the design of future clinical studies evaluating CD47-targeted combinations and treatment sequencing in HNSCC. These data can help further optimize how we enhance antitumor activity while reducing tumor evasion mechanisms in both HPV-positive and negative HNSCC.

## Methods

All the animal studies were approved by the University of California San Diego (UCSD) Institutional Animal Care and Use Committee (IACUC, protocol #S16200); all experiments adhere to all relevant ethical regulations for animal testing and research. All researchers involved in animal studies received the appropriate training from the UCSD IACUC.

### Cell lines and tissue culture

The 4MOSC (4MOSC1, 4MOSC2, 4MOSC1 CD47 KO, 4MOSC1 Cas9 KO, 4MOSC1 CD47 KO Cas9 KO) syngeneic mouse HNSCC cells harboring a human tobacco-related mutanome and genomic landscape were developed and described for use in immunotherapy studies in a prior report [23]. 4MOSC cells were cultured in Defined Keratinocyte-SFM medium (ThermoFisher, 10744019) supplemented with EGF Recombinant Mouse Protein (5 ng/ml) (ThermoFisher, PMG8044), Cholera Toxin (50 pM) (Sigma-Aldrich, C8052), and 1% antibiotic/antimycotic solution. 293T cells (ATCC CRL-3216) were cultured in Dulbecco's Modified Eagle's Medium (DMEM, ThermoFisher, 21-041-025) supplemented with 10% fetal bovine serum, 2 mM L-glutamine (ATCC 30–2214) and 1% antibiotic/antimycotic solution. All cells were cultured at 37 °C in the presence of 5% $CO_2$.

### Cloning of pLenti-eGFP-LucOS

The full-length coding sequence of LUC-OS flanked by attbB1/2 recombination site was amplified from the Lenti-LucOS (22777) using the LUC-OS-F (5′-GGGGACAAGTTTGTACA AAAAGCAGGCTTAATGGAAGACGCCAAAAACATA-3′) and LUC-OS-R (5′-GGGG ACCACTTTGTACAAGAAAGCTGGGTTTTACAAGTCCTCttCAGAAAT-3′) primer. The purified PCR product was incorporated into the pDONR221 vector via a BP Reaction and subsequently introduced into the pLenti-CMV-GFP-DEST (19732) through an LR reaction.

### Generation of stable GFP-Luc and eGFP-LucOS expressing 4MOSC1 and 4MOSC2 cell line

For lentivirus production, 293T cells were plated in a poly-D-lysine–coated 15-cm dish and, 16 h later, transfected with 30 mg pHAGE PGK-GFP-IRES-LUC-W or pLenti-eGFP-LucOS, 3 mg VSV-G, 1.5 μg Tat1b, 1.5 μg Rev1b, and 1.5 μg Gag/

Pol using 25.2 µL P3000 reagent and 25.2 µL of Lipofectamine 3000 transfection reagent, and media was refreshed 16 h post-transfection. At 48 and 72 h, virus-containing media was collected, filtered througfh a low protein binding filter unit (PVDF, 0.45 µm, Sigma-Aldrich), and stored at 4 °C for up to 5 days prior to use. Lentivirus suspension was concentrated using Lenti-X concentrator per manufacturer standardized protocol (Takara Bio). Subsequently, 4MOSC1/2 cells were plated in a collagen-coated 6-well plate. At 16 h, seeded cells were transduced using 200 µL of concentrated virus in 2 mL keratinocyte-defined serum-free media and 4 µg/mL polybrene, and the plate was immediately centrifuged for 15 min at 450 × g. GFP expression was validated by fluorescent microscopy and flow cytometry. Transduced 4MOSC1/2 cells were sorted by FACS for viability and GFP-positivity using a FACS-Aria Cell Sorter (BD Biosciences).

**Generation CD47 knockout 4MOSC1 cell line using selective CRISPR antigen removal lentiviral vector system**

Lentiviral vectors pSCAR_Cas9, IDLV-Cre, and pSCAR_sgRNA (Cd47 sgRNA: CCACATTACGGACGATGCAA) were generated and obtained from a previous study [24]. 4MOSC1 cell line was first infected with pSCAR_Cas9 in media containing polybrene (Millipore Sigma #107689); after 48 hours they were selected with blasticidin (InvivoGen #ant-bl-1) for 7 days. Cas9 expression by 4MOSC1 was verified via western blot using primary antibodies (α-tubulin: Cell Signaling Technology #3873S; Cas9: Cell Signaling Technology #14697), HRP-conjugated secondary antibodies, and enhanced chemiluminescence (S1 Fig 1B). Cas9-expressing cells were then infected with lentivirus pSCAR_sgRNA and after 48 hours, they underwent selection with blasticidin for 3 days. Following pSCAR_sgRNA transduction, cells were cultured in blasticidin for a total of 10 days to allow genome editing while maintaining high expression of SCAR vectors. At the end of 10 days, cells were infected with IDLV-Cre in media containing polybrene and blasticidin. Following IDLV-Cre transduction (by the same process as other lentivirus infections), cells were sorted by FACS for viability and GFP and mKatie negativity using FACS-Aria Cell Sorter (BD Biosciences).

**In vivo mouse models and analysis**

All the animal studies using HNSCC orthotropic implantation studies were approved by the UCSD Institutional Animal Care and Use Committee (IACUC), with Animal Study Proposal (ASP) protocol #S16200; and all experiments adhere to all relevant ethical regulations for animal testing and research. All mice were obtained from Jackson Laboratories (San Diego, CA). Mice at UCSD Moores Cancer Center are housed in individually ventilated and micro-isolator cages supplied with acidified water and fed 5053 Irradiated Picolab Rodent Diet 20. The temperature for laboratory mice in this facility is mandated to be between 18 and 23 °C with 40–60% humidity. The vivarium is maintained in a 12-h light/dark cycle. All personnel were required to wear scrubs and/or lab coat, mask, hair net, dedicated shoes, and disposable gloves upon entering the animal rooms. WT C57Bl/6 female mice were obtained from Jackson Laboratories (San Diego, CA). Mice were randomly assigned to each treatment group after confirmation of the tumor implantation before treatment. Investigators were not blinded during outcome assessment.

**Orthotopic tumor modeling**

For orthotopic implantation, 4MOSC1 cells were transplanted (1 million per tumor) into the oral cavity's buccal mucosa of female C57Bl/6 mice (4–6 weeks old). Similarly, 4MOSC2 cells were transplanted (0.5 million per tumor) into the same location of female C57Bl/6 mice (4–6 weeks old). Mice were anesthetized using isoflurane inhalation mixed with oxygen during all procedures and tumor monitoring. For drug treatment, the mice received intraperitoneal injections (ip) of either anti-PD-1 monoclonal antibody (BioXCell, BE0033−2) or anti-CD47 fusion protein (ALX301). The mice were monitored twice weekly for weight, tumor size/growth, activity levels, and overall appearance. They were sacrificed at the specified time points or when they reached humane endpoints (buccal tumors >1 cm or ulcerated), as per ASP guidelines. Tissues were collected for histological, immunohistochemical, or flow cytometric analysis. The maximum tumor size/burden

allowed by our institutional review board was not exceeded. Once mice reached the designated time points, they were euthanized the same day with carbon dioxide, followed by cervical dislocation to ensure euthanasia. 450 mice were used for this project, and no animals died before meeting the criteria for euthanasia. Each experiment involving animals lasted no more than three weeks post-orthotopic implantation.

## Lymphatics mapping

Lymphatic mapping was done by injecting lymphazurin into the buccal mucosa and visualizing the blue dye as it drained to the cervical draining lymph node. Lymphazurin was prepared by dissolving 10 mg of isosulfan blue (MedChemExpress, HY-107967) reconstituted in 353µl of DMSO (Millipore Sigma, D8418), 1.412 ml of PEG300 (Millipore Sigma, 807484), 176.5µl of TWEEN-80 (Millipore Sigma, P5188), and 1.589 ml of sterile saline.

## Radiation

For the tumor-directed Radiation Therapy (tdRT), mice were treated with one dose of 4Gy stereotactic body radiotherapy (SBRT) focused on the buccal mucosa containing the tumor. tdRT was done on day 6 post transplant of the tumor prior to ICI treatments.

## Preparation of ICI treatments

All anti-PD1 treatments were prepared at a dose of 10 mg/kg and mice were treated on days 6 and 8. All ALX301 treatments were prepared at a dose of 30 mg/kg and mice were treated on days 6, 10, 14, and 18 except for flow cytometry experiments where takedown occurred earlier. The vehicle used for the control was PBS.

## Flow cytometry analysis

Immunophenotyping of tumor-bearing mice was performed according to the STAR protocol with the following modifications [37]. Briefly, tumor tissue, sentinel lymph nodes, and non-sentinel lymph nodes were mapped and harvested using lymphazurin. To obtain a single-cell suspension, tissues were passed through 70 µm cell strainers and treated with ACK buffer for red blood cell lysis. The single-cell suspension was then stained with Zombie NIR™ Fixable Viability Kit (BioLegend). After Fc blocking with TruStain FcX™ (anti-mouse CD16/32 antibody) (BioLegend), the cells were stained with fluorochrome-conjugated antibodies in Brilliant Stain Buffer (BD Biosciences). The cells were fixed with 2% paraformaldehyde and analyzed on Agilent NovoCyte Advanteon (Agilent Technologies) with standard lasers and optical filters. Data was analyzed using FlowJo (FlowJo, LLC). The following fluorochrome-conjugated antibodies (clone, dilution) were used in this study: CD45.2 (104, 1:200), CD8α (53-6.7, 1:200), NK1.1 (PK136, 1:100), CD11c (N418, 1:200), F4/80 (BM8, 1:100), CD197/CCR7 (4B12, 1:50), CD69 (H1.2F3, 1:100), CD86 (GL-1, 1:100) from BioLegend, and I-A/I-E (2G9, 1:500) from BD Biosciences.

## TCR sequencing

TCR sequencing was prepared by harvesting one tumor and one SLN from one mouse of each treatment group. Immediately after harvesting, RNA was prepared using QIAGEN RNA extraction kit. RNA samples were sent to CD Genomics for sequencing.

## Statistics

Data analysis was performed using GraphPad Prism 10 for MacOS. Differences in experimental groups were analyzed using independent t-tests, one-way ANOVA with multiple comparisons, or two-way ANOVA with multiple comparisons. Survival Analysis was performed using the Kaplan-Meier method and Survival Analysis. All data are reported as mean±SEM.

All alpha thresholds were 0.05 and all confidence intervals were 95%. For P-values in figures, a P value of less than 0.0001 is denoted as ****, a P value between 0.0001 and 0.001 is denoted as ***, a P value between 0.001 and 0.01 is denoted as **, and a P value between 0.01 and 0.05 is denoted as *. Samples used in experiments were biological replicates.

## Supporting information

**S1 Fig. (A) pSCAR_sgRNA (Cd47 sgRNA: CCACATTACGGACGATGCAA) used in the knockout of the Cas9 and CD47 represented in Fig 1C [24].** (B) Verification of Cas9 expression in 4MOSC1 using western blot. (C) Kaplan Meier curve demonstrating the survivability from the experiment depicted in Fig 2C (n = 10 for each arm, Control vs. anti-PD1 + tdRT p = 0.0505, Control vs. anti-PD1 + ALX301 p = 0.0114*, Control vs. anti-PD1 + ALX301 + tdRT p = 0.016*) where with the treatment of ALX301, there is a statistically significant increase in survivability. p values were calculated using a Log-rank test. (D) Representative tumor growth kinetics of mice with 4MOSC1 tumors treated with vehicle (control) vs. ALX301. Data are presented as mean ± SEM (n = 8 for control, n = 7 for ALX301). (E) Representative tumor growth kinetics of mice with 4MOSC2 tumors treated with vehicle (control), anti-PD1 + tdRT, anti-PD1 + ALX301, and anti-PD1 + ALX301 + tdRT. Data are presented as mean ± SEM (For control: n = 10 up to day 6, n = 9 on day 8. One mouse was omitted due to succumbing to disease. n = 10 for all other groups). (F) Representative tumor growth kinetics of mice with 4MOSC2 tumors treated with vehicle (control), ALX301, anti-PD1, ALX301 + anti-PD1, anti-PD1 + tdRT, and ALX301 + anti-PD1 + tdRT. Data are presented as mean ± SEM (n = 10 for control, n = 8 for all other groups). (G) Representative tumor growth kinetics of mice with 4MOSC2 tumors treated with vehicle (control), ALX301, tdRT, and ALX301 + tdRT (p = 0.9996 for Control vs. ALX301, p = 0.1841 for Control vs. tdRT, *p = 0.0188 for Control vs. ALX301 + tdRT, p = 0.2671 for ALX301 vs. tdRT, *p = 0.0436 for ALX301 vs. ALX301 + tdRT, p = 0.9341 for tdRT vs. ALX301 + tdRT). Data are presented as mean ± SEM; p values were calculated using ordinary two-way ANOVA with Tukey's post-hoc (For control: n = 8 up to day 10, n = 6 on day 12, n = 5 on days 14–18. For ALX301: n = 8 up to day 10, n = 7 on day 12, n = 4 on days 14–18. For tdRT: n = 7 up to day 10, n = 5 on day 12, n = 4 on days 14–18. For ALX301 + tdRT: n = 7 up to day 18). Mice were omitted due to succumbing to disease.
(TIF)

**S1 Raw Images. Original western blot image used to verify cas9 expression in the 4MOSC1 cell line.**
(PDF)

**S2 File. Reporting of statistical results. summary of statistical analyses performed in this study.**
(DOCX)

## Acknowledgments

We thank ALX Oncology for providing the ALX301 (anti-CD47) reagent used in this study. Cartoon renderings were created with the BioRender online platform (BioRender.com).

## Author contributions

**Conceptualization:** Abdula Monther, Riyam Al-Msari, Silvio Gutkind, Joseph Califano.

**Data curation:** Abdula Monther.

**Formal analysis:** Abdula Monther, Riyam Al-Msari, Santiago Fassardi, Sayuri Miyauchi.

**Funding acquisition:** Andrew Sharabi, Joseph Califano.

**Investigation:** Abdula Monther, Riyam Al-Msari, Robert Saddawi-Konefka, Cynthia Tang, Chad Philips, Prakriti Sen, Pardis Mohammadzadeh, Kelsey Decker, Sayuri Miyauchi, Souvick Roy, Riley Jones, Xingyu Wu.

**Methodology:** Abdula Monther, Riyam Al-Msari, Robert Saddawi-Konefka, Joseph Califano.

**Project administration:** Joseph Califano.

**Resources:** Joseph Califano.

**Supervision:** Joseph Califano.

**Validation:** Abdula Monther.

**Visualization:** Abdula Monther, Riyam Al-Msari, Santiago Fassardi, Xingyu Wu.

**Writing – original draft:** Abdula Monther, Riyam Al-Msari.

**Writing – review & editing:** Abdula Monther, Robert Saddawi-Konefka, Silvio Gutkind, Andrew Sharabi, Joseph Califano.

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
