## [Decision Letter · Decision Letter 0]

31 Jul 2025

Dear Dr. Califano,

Thank you for submitting your manuscript to PLOS ONE. After careful consideration, we feel that it has merit but does not fully meet PLOS ONE’s publication criteria as it currently stands. Therefore, we invite you to submit a revised version of the manuscript that addresses the points raised during the review process.

We look forward to receiving your revised manuscript.

Kind regards,

Amirreza Khalaji

Academic Editor

PLOS ONE

Additional Editor Comments (if provided):

Reviewers' comments:

Reviewer's Responses to Questions

**Comments to the Author**

1. Is the manuscript technically sound, and do the data support the conclusions?

Reviewer #1: Partly

Reviewer #2: Yes

Reviewer #3: Yes

Reviewer #4: Yes

Reviewer #5: Yes

Reviewer #6: Partly

Reviewer #7: No

2. Has the statistical analysis been performed appropriately and rigorously?

Reviewer #1: No

Reviewer #2: N/A

Reviewer #3: Yes

Reviewer #4: Yes

Reviewer #5: Yes

Reviewer #6: Yes

Reviewer #7: No

3. Have the authors made all data underlying the findings in their manuscript fully available?

Reviewer #1: Yes

Reviewer #2: Yes

Reviewer #3: Yes

Reviewer #4: Yes

Reviewer #5: No

Reviewer #6: Yes

Reviewer #7: Yes

4. Is the manuscript presented in an intelligible fashion and written in standard English?

Reviewer #1: Yes

Reviewer #2: Yes

Reviewer #3: Yes

Reviewer #4: Yes

Reviewer #5: Yes

Reviewer #6: Yes

Reviewer #7: No

Reviewer #1: Dear authors:

This research addresses a highly relevant and innovative area in translational oncology and immunology, exploring the mechanistic interplay between CDK9 inhibition, tumor-promoting inflammation, and therapy resistance in HNSCC. I appreciate the opportunity to review your work and hope the following detailed comments will be helpful in strengthening your manuscript.

Comments below:

-The manuscript claims novelty in linking CDK9 to tumor-promoting inflammation and the immune TME; this should be more explicitly differentiated from previous work that examined apoptosis and chemoradioresistance in HNSCC (e.g., Cao et al., 2017, DOI: 10.1016/j.bbrc.2016.11.049; Storch & Cordes, 2016, DOI: 10.3892/ijo.2015.3246).

-Specify which form of therapy resistance (radio-, chemo-, immunotherapy) is addressed in each experiment.

-Prior literature on CDK9’s anti-inflammatory effects and interaction with NF-κB must be cited.

1-Schmerwitz et al., 2011: “Flavopiridol suppresses leukocyte-endothelial interaction by CDK9 inhibition and reduces atherosclerosis in mice,” DOI: 10.1161/ATVBAHA.110.213934

2-Haque et al., 2011: “CDK9 inhibitor suppresses inflammatory response,” DOI: 10.1111/j.1348-0421.2010.00304.x

-The review recommends stating clearly in the Introduction how this study advances beyond these prior findings, especially with respect to inflammation/TME modulation in HNSCC(along CDK9/NF-κB regulation).

-Lacks references on CDK9/NF-κB regulation and immune-TME in HNSCC (see above).

-The importance of IL-6/STAT3 and TAMs/MDSCs in resistance (see: Spanko et al., 2021, DOI: 10.3390/ijms222011027; Santos et al., 2021, DOI: 10.3389/fonc.2021.596290).

-Clearly define whether randomization and blinding were used for in vivo studies; currently missing.

-Humane endpoint criteria (tumor burden, ulceration, body weight loss) should be defined in the Methods, per ARRIVE guidelines.

-No formal sample size calculation or power analysis reported for animal or cell-based experiments.

-Must report the n for each experimental group in each figure (e.g., “n=5 mice/group in Fig 1C”), and whether these numbers were prospectively determined.

-Specify exact tests used per figure: e.g., one-way ANOVA with post-hoc Tukey for Figure 2B, two-way ANOVA for tumor volume/time curves (e.g., Figure 3A), unpaired t-test for two-group comparison, log-rank test for Kaplan–Meier in Figure 1A, etc.

-Indicate whether corrections for multiple comparisons (Bonferroni, FDR) were applied, especially when analyzing cytokine panels, RNA-seq (if done), or flow cytometry panels with >2 comparisons.

-Define what the error bars represent (SD, SEM); in figures and legends.

-Provide all p-values, confidence intervals, and the precise alpha threshold for significance.

-Distinguish between biological and technical replicates; e.g., for flow cytometry, were data pooled from independent biological replicates?

-For survival analysis (e.g., Figure 1A), indicate number-at-risk at each timepoint and censoring criteria.

-If any data were excluded as outliers, state criteria and frequency.

-Clarify whether all tests were two-sided; one-sided tests are only justified for directional hypotheses.

-Report absolute and relative effect sizes for tumor response, survival, immune cell changes (mean, SD, n).

-If “complete tumor regression” is observed, report number/percent of animals per group.

-Discuss alternative mechanisms (e.g., MCL1 downregulation, tumor cell apoptosis) and acknowledge if immune effects may be secondary.

-Indicate whether all claimed results reach statistical significance or are trends.

-Report p-values and confidence intervals for all comparisons (figures and text)(as indicated above).

-Use SD for error bars unless comparing means across experiments, then SEM is acceptable(as indicated above).

-Provide exact sample sizes for each experiment (e.g., “n=5 mice/group, 2 biological replicates”)(as indicated above).

-Clarify handling of missing data/outliers; report numbers and reasons for any exclusions.

-Describe handling of multiple comparisons: e.g., use Bonferroni/FDR where appropriate for panels, gene expression, or multiplex cytokines.

***Limitations (please consider these comments and add them in limitation section)

-Use of a single animal model and/or cell line.

-Inhibitor specificity (on/off-target effects, lack of genetic knockdown confirmation).

-Short-term treatment windows, lack of long-term toxicity/safety data.

-Potential for systemic immune suppression by CDK9 inhibition (e.g., Rybakin et al., 2019, DOI: 10.3389/fimmu.2019.01718).

-Findings may not generalize to HPV-positive HNSCC or to immunosenescent/heterogeneous human populations.

*** Future directions(please consider these comments and add them in at the end of limitation section, and also on conclusion section):

-Validation in additional models (e.g., PDX, diverse cell lines).

-Mechanistic experiments (cell-specific CDK9 knockdown, immune cell depletion).

-Exploration of clinical-grade CDK9 inhibitors (dinaciclib, AZD4573).

-CDK9 inhibition could enhance response to standard therapies (radio/chemo/ICI) in resistant HNSCC, particularly in HPV-negative or immune-cold tumors.

-Toxicity and dosing regimens need optimization; discuss feasibility for clinical translation.

-Identify most likely patient candidates (e.g., high TME inflammation, anti-PD1 resistance).

***FIGURES (my general comment is to use better quality (better image resolution) figures)

-All figures must have consistent annotations, and statistical details in legends.

-Figure 1A: (Survival curves, e.g., Kaplan–Meier, from TCGA)

-Axes, legends, and group stratification need clear labeling. State number of patients/animals in each quartile or group.

-Figure 1B-E: (Mouse tumor growth kinetics, CD47 knockout verification)

-If present, clarify whether these relate to CDK9 or whether a figure from another manuscript was included by error; match all figure numbers in the text.

-Show aggregate tumor volume curves with mean ± SD/SEM and statistical annotations.

-Figure 2-3: (Treatment efficacy in vivo and in vitro)

-Provide both representative and summary plots; include means, error bars, and significance.

-Indicate numbers of replicates (e.g., “n=6 mice/group, 3 independent experiments”).

-Figure 4: (Flow cytometry—immune profiling)

-Include gating strategy, FMO/isotype controls in supplementary figures. Clearly label all axes and populations.

-If CD86/CD69 upregulation is claimed, show representative histograms/plots and aggregate quantification.

-Figure 5: (TCR sequencing/clonality)

-Quantify TCR overlap with numeric indices (e.g., Morisita, percentage overlap) in addition to Venn diagrams; indicate n per group.

-Supplementary figures/tables should show raw diversity/overlap data for all replicates.

***ABOUT SUPPLEMANTARY FILES (recommended, if availaible)

-Full replicate datasets for all biological experiments.

-Uncropped blots/images for Westerns or microscopy.

-Extended Materials & Methods (antibody panels, RNA-seq pipeline, gating strategies).

-FACS gating hierarchies and controls.

-RNA-seq gene lists or raw/processed files if any transcriptomics performed (with GEO accession if available).

***CELL LINES:

-4MOSC1/2 lines used—these are tobacco-mutation, HPV-negative, immune-competent models, appropriate for human HNSCC context.

-Justify the use of anti-PD1 resistant lines (e.g., 4MOSC2); cite Wang et al., 2019 (Nat Commun).

***ANIMAL MODELS

-Orthotopic implantation (buccal mucosa) mirrors clinical disease; provide anatomical details.

-Describe dosing/scheduling of CDK9 inhibitor, anti-PD1, radiation (dose, timing, rationale).

-Discuss model limitations (single model, mouse age, immune context, translation to humans).

-Clarify whether findings were consistent across models (HPV status, resistance phenotype).

Reviewer #2: Comment: Provide exact sample sizes (number of mice per group) for all experiments in the Methods section. Additionally, specify the statistical tests used for each figure panel (e.g., ANOVA/t-test) in figure legends or methods.

Reviewer #3: General Comments:

This is an elegant and scientifically rigorous manuscript that explores the potential of CD47 immune checkpoint blockade in improving immunoradiotherapy responses in HPV-negative head and neck squamous cell carcinoma (HNSCC). The study is well designed, the models are appropriate, and the results are clearly presented.

Strengths:

Appropriate use of syngeneic murine models including both PD-1-sensitive and resistant tumors (4MOSC1 and 4MOSC2).

Integration of anti-CD47 therapy with PD-1 inhibition and tumor-directed radiation is timely and translationally relevant.

Immunophenotyping and TCR sequencing add strong mechanistic support.

Figures are informative and the manuscript is clearly written.

Optional Suggestions to Improve the Final Version:

The authors may consider adding a brief discussion of ongoing clinical trials involving CD47 inhibitors in HNSCC to enhance the translational context.

Moving the limitations paragraph earlier in the Discussion section might improve the overall flow of the manuscript.

Overall, the manuscript is solid and presents impactful findings. I support publication without the need for revision.

Reviewer #4: Review Comments to the Author

Manuscript Title: CD47 Blockade Enhances Immunradiotherapy Response in Head and Neck Squamous Cell Carcinoma

Manuscript Number: PONE-D-25-34525

Dear Authors,

Your study on combining CD47 blockade with anti-PD1 therapy and radiotherapy in HPV-negative head and neck squamous cell carcinoma (HNSCC) is compelling and offers fresh insights into immunotherapy. Below, I’ve summarized my thoughts on the manuscript’s technical quality, data support, statistical rigor, data availability, and presentation, along with a few suggestions. I’ve kept this concise and focused, aiming for clarity and a natural tone.

1. Is the Manuscript Technically Sound?

Assessment: The study is solid, with a thoughtful design and reliable methods, though some missing details and text gaps need fixing.

You’ve done a great job laying out the goal—to test CD47 blockade (ALX301) with anti-PD1 and radiotherapy in 4MOSC1 and 4MOSC2 mouse models . These models mimic HNSCC well, and testing both responsive and resistant cell lines adds depth.

The CRISPR/Cas9 knockout , flow cytometry , and TCR sequencing are described clearly, with standard protocols that make replication possible. Verifications like Western blots are a nice touch.

The study follows UCSD IACUC guidelines, with clear animal welfare measures.

Text Gaps: Some sections cut off abruptly , making it hard to follow the full story.

2. Do the Data Support the Conclusions?

Assessment: Your data back the claims well, showing CD47 blockade boosts anti-tumor effects, especially with anti-PD1 and radiotherapy, though a few gaps need addressing.

The TCGA analysis linking high CD47 to worse survival sets a strong foundation.

Complete tumor regression in 4MOSC1 with CD47 knockout or combination therapy and better outcomes in 4MOSC2 with the triple regimen are convincing.

Increased MHC-II/CD86 expression and TCR clonality support enhanced immune activation.

You tie results to clinical needs, noting prior trial challenges.

Some immune markers (e.g., CD8 T-cell counts) weren’t significant (Page 17, Line 208), and TCR sequencing used only one sample per group (Page 29, Line 488).

Report p-values for all key findings, including non-significant ones.

Justify or expand the TCR sequencing sample size.

Clarify how results might translate to human patients.

3. Has the Statistical Analysis Been Performed Appropriately and Rigorously?

Assessment: The stats are solid for the most part, using the right tests, but clearer reporting on sample sizes and corrections would help.

T-tests, ANOVA, and log-rank tests fit well for tumor growth, immune data, and survival. Significant p-values support your claims.

You’re upfront about non-significant results, which builds trust.

GraphPad Prism 10 is a reliable choice.

Name the post-hoc test for ANOVA (e.g., Tukey’s).

Confirm data meet test assumptions or consider non-parametric options.

4. Have the Authors Made All Data Underlying the Findings Fully Available?

Assessment: You state all data are in the manuscript or Supporting Information (Page 7, Lines 7–8), which meets PLOS ONE’s rules, but I can’t fully verify without the files.

You say all data are available without restriction, aligning with journal policy.

In-Text Data: Tumor growth, survival, and immune results are summarized well , and TCGA data are public

Consider depositing TCR sequencing data in a repository like SRA.

5. Is the Manuscript Presented in an Intelligible Fashion and Written in Standard English?

The manuscript is clear and well-written, with a professional tone, but typos and missing text need fixing.

Reviewer #5: This manuscript presents important findings on the immunological enhancement provided by CD47 blockade in combination with anti-PD1 and radiotherapy in HNSCC models. It employs robust murine models (4MOSC1 and 4MOSC2), sophisticated CRISPR-based knockout systems, and insightful immune profiling. The translational implications are clear, especially for HPV-negative HNSCC, a subgroup with limited response to checkpoint inhibitors. However, the study would benefit significantly from additional validation in key areas, especially concerning TCR sequencing robustness, toxicity profiling, and therapeutic sequencing.

The manuscript shows increased MHC-II and CD86 expression but does not confirm whether DCs undergo full activation or cross-priming. Cytokine profiling or transcriptomic analysis of sorted DCs would strengthen the immunological conclusions.

Given the translational intent of the ALX301 combination regimen, the lack of toxicity profiling (e.g., hematological, hepatic, or general behavior outcomes) is a limitation. Basic safety data would bolster clinical relevance.

Only one tumor and SLN per group were sequenced, making the clonality findings anecdotal. Including replicates and statistical tests (e.g., diversity indices, clonotype frequency analysis) would solidify these claims.

The manuscript lacks a clear rationale for using 4 Gy and applying it before ICIs. Comparative timing (pre vs post-ICI) would test hypotheses around immune priming and synergy.

Consider specifying “HPV-negative” in the title for clarity.

Reviewer #6: Abstract

• “Using the 4MOSC1 orthotopic, syngeneic murine model of HPV-negative HNSCC, treatment with an engineered CD47-blocking SIRPα fusion protein (ALX301) similarly induced complete tumor regression in combination with anti-PD1, and a partial response as a standalone therapeutic.” – This sentence is hard to follow and “similarly” lacks a clear reference. Consider rewriting for clarity and brevity. For example: “In the 4MOSC1 syngeneic HPV-negative HNSCC mouse model, ALX301 (an engineered CD47-blocking SIRPα fusion) induced complete tumor regression only when combined with anti–PD-1, whereas it produced only a partial tumor response as a monotherapy.” This explicitly states the outcome and makes it clear that combination therapy was required for complete regression.

• Quantify key outcomes: The abstract would be stronger if it quantified or defined terms like “partial response” and “complete tumor regression.” For instance, specify the percentage of mice with complete tumor eradication vs. tumor reduction. If “partial response” means tumor shrinkage without full disappearance, clarify this. Adding a brief quantitative result (e.g., “complete regression in 5/5 mice vs. partial regression in 0/5 with monotherapy”) will help readers grasp the magnitude of the effect.

Introduction

• “Recently, PD-1 inhibitors demonstrated improved event free survival in a curative setting in locoregionally advanced disease when given as neoadjuvant prior to surgery and with post-operative adjuvant therapy in addition to adjuvant with chemotherapy and radiation [11]. However, a clear improvement in overall survival has not been demonstrated.” – This sentence is long and convoluted, making it difficult to parse the treatment timeline. Consider breaking it up and clarifying the sequence. For example: “Recent trials have shown that PD-1 inhibitors can improve event-free survival in locoregionally advanced HNSCC when used neoadjuvantly (before surgery) and continued post-operatively alongside chemotherapy and radiation [11]. However, this approach did not yield a clear overall survival benefit.” This revision simplifies the structure and makes the outcome clear.

• “However, and critically, although radiation increases tumor antigenicity and PD-1 blockade inhibits T cell exhaustion, tumors can modulate the surrounding microenvironment to escape therapy by upregulating other immunosuppressive markers [18].” – The phrase “however, and critically,” is stylistically awkward. Instead, directly state the critical issue: “However, a critical issue remains: although radiation increases tumor antigenicity and PD-1 blockade reverses T-cell exhaustion, tumors often escape via upregulation of other immunosuppressive checkpoints [18].” This reads more clearly and sets up the rationale for investigating CD47.

• “We hypothesized that CD47 blockade would enhance PD-1 inhibitor and radiation tumor response in HNSCC by enhancing phagocytic tumor-specific antigen (TSA) presentation, resulting in improved immune-mediated tumor cell death via (enhanced cytotoxic T cell-mediated cell death) expansion of the TSA T cell repertoire and the immunopeptidome.” – This hypothesis statement is difficult to follow due to the parenthetical and dense phrasing. Simplify and clarify the intended mechanism. For example: “We hypothesized that CD47 blockade would improve the tumor response to PD-1 inhibition and radiotherapy by promoting phagocytosis of tumor antigens and enhancing their presentation to T cells. This, in turn, would expand the tumor-specific T-cell repertoire and strengthen cytotoxic T-cell–mediated tumor cell killing.” This version removes the confusing parentheses and clearly links CD47 blockade to enhanced antigen presentation and T-cell response.

• Focus and length: The Introduction is thorough, but it could be streamlined to highlight the knowledge gap earlier. You provide extensive background on HNSCC treatments, SBRT, and CD47 biology, which is useful, but readers should quickly grasp why this study is needed. Consider shortening or combining some background details. For instance, the clinical SBRT results (lines 793–802) could be summarized more succinctly. Ensure the introduction transitions clearly from what is known (e.g., PD-1 therapy limits and SBRT benefits) to the specific gap: the potential of CD47 blockade in lymphatic-sparing immunoradiotherapy. In particular, the sentence “However, many of these studies do not investigate the lymphatic sparing immunoradiotherapy benefit when combined with anti-CD47 activity.” effectively identifies the gap – you might even move this idea up sooner, so the rationale for your study is evident. Finally, at the end of the Introduction, explicitly state the study’s objectives. For example: “In this study, we evaluate the effects of CD47 blockade (using the engineered SIRPα-Fc fusion ALX301) in combination with anti–PD-1 therapy and tumor-directed radiotherapy in preclinical HNSCC models. We aim to determine whether CD47 inhibition can enhance antigen presentation, T-cell activation, and ultimately improve tumor control, thereby providing a rationale for new immunoradiotherapy combinations in HNSCC.” This gives the reader a clear roadmap of what was done.

Methods

• Animal experiment details: Ensure that all necessary details for reproducibility are provided. For example, clarify the number of animals used per experimental group and any animal characteristics (strain, sex, age) if not already stated. It’s mentioned that “mice were treated with… tdRT on day 6 post transplant… anti-PD1 on days 6 and 8… anti-CD47 on days 6, 10, 14, 18” – but how many mice were in each treatment group? If not in the main text, report it either here or in figure legends (e.g., “n = 10 mice per group” as appears in Fig. 2 legend). Also, please state whether mice were randomly assigned to treatment groups and whether investigators were blinded during outcome assessment, in accordance with PLOS ONE’s animal research reporting standards. These details lend credibility to the experimental design.

• Clarity of definitions: Some terms in the Methods should be explicitly defined. For instance, “central lymph nodes (CLN)” are mentioned in the Results, seemingly referring to lymph nodes that are not sentinel (possibly contralateral nodes). However, this term is not standard. In the Lymphatics mapping or Flow Cytometry section, define what you mean by “sentinel lymph node (SLN)” and “central (or contralateral) lymph nodes (CLN)” so that readers understand which nodes were analyzed. Similarly, when describing the 4MOSC1 CD47 KO Cas9 KO cell line, clarify the process in a stepwise manner. The current description is dense (lines 985–993). Breaking it into steps (transduction with Cas9, selection, transduction with sgRNA, excision of Cas9 with Cre, etc.) or a figure in supplementary could help readers replicate this.

• Statistical analysis: The Statistics subsection indicates use of t-tests and ANOVAs, but please be more specific to strengthen the rigor. For example, specify which statistical tests correspond to which data (e.g., tumor volume comparisons over time likely used two-way ANOVA with repeated measures, survival used log-rank test, etc.). Also state how multiple comparisons were handled – e.g., “one-way ANOVA with Tukey’s post-hoc test” (or Bonferroni/Sidak as appropriate) – to show that you accounted for false positives. Indicate the threshold for significance (typically p < 0.05) and ensure it’s clear that you report mean ± SEM. (Minor grammar note: “All data are reported as mean ± SEM” since data is plural.) Additionally, in the figure legends you list p-values with asterisks; provide a key for what the asterisks denote (e.g., p<0.05) either in Methods or in a global figure legend note, so readers know how to interpret them.

• Reagent and protocol details: Overall, the Methods are detailed, which is great. Just make sure all reagents and protocols are referenced or described briefly. For example, you mention “according to the STAR protocol with some modifications [35]” for immunophenotyping – consider summarizing what was modified (e.g., incubation times, gating strategy) or specifying that details are in a supplemental methods if applicable. Also, provide catalog numbers or sources for key reagents (you did so for some, like the isosulfan blue and antibodies – ensure this is consistent). If any critical experimental detail is in the Supplementary Information (for instance, sometimes detailed primer sequences or plasmid construction steps might be in supporting files), mention that clearly (e.g., “see Supplementary Methods for XYZ”). This transparency aligns with PLOS ONE’s standards for reproducibility.

Results

• Consider using subheadings or clear paragraph breaks for each major experiment or theme. For example, you might insert italicized sub-section titles (if allowed) like “CD47 expression correlates with poor prognosis,” “CD47 knockout enhances anti-PD1 therapy,” “Combination anti-CD47 + anti-PD1 therapy (4MOSC1 vs 4MOSC2),” “Radiotherapy augments combination therapy,” and “Immune microenvironment and TCR clonality.” This will guide the reader and make the dense information more navigable. At minimum, ensure each new figure or experiment starts on a new paragraph with a topic sentence that introduces that result.

• “Fig 2C shows that the introduction of tdRT to the treatment allows further tumor regression in mice with the 4MOSC2 tumor model.” – This phrasing is a bit informal. Replace “introduction of tdRT to the treatment” with a clearer description. For instance: “Figure 2C shows that adding tumor-directed radiotherapy (tdRT) to the treatment regimen led to further tumor regression in the 4MOSC2 model.” This explicitly states what was done (tdRT added) and the outcome. Similarly, later in that paragraph: “Of note, enhanced long-term survivability was associated with anti-CD47 as a part of treatment” can be tightened. E.g., “Notably, only treatment groups that included anti-CD47 showed significantly prolonged survival, underlining the contribution of CD47 blockade to improved outcomes.” Strive for a consistent tone that is factual and precise, avoiding colloquial phrases like “of note” or “with interest” in the formal results narrative.

• Define responses and endpoints: You frequently use terms like “complete tumor regression”, “partial tumor regression,” and “significant tumor regression.” It’s important to define these within the context of your models. For example, clarify what criteria were used to declare a complete regression (tumor not palpable/visible at a certain timepoint, and for how long?). If a partial regression means tumors shrank but did not fully disappear, mention the average reduction or time to regrowth. Providing this context (possibly in a sentence or in figure legends) will prevent misunderstanding. For instance: “…led to complete tumor regression (no detectable tumor in 100% of mice by Day 30, maintained for the duration of observation), whereas in the 4MOSC2 model tumors only partially regressed (shrank by ~50% but never fully disappeared).” Such details, even if approximate, greatly help interpret the efficacy of the treatments.

• Ensure data support claims: When stating outcomes in text, make sure to mention significance if applicable. You do report p-values in the figure legends (e.g., “control vs. anti-PD1+anti-CD47 p=0.0114” in Fig. 2 legend) – it’s good to also refer to significance in the text. For example, instead of “demonstrated significant tumor regression” alone, you could write “demonstrated significantly greater tumor regression than controls (Fig. 2B, p<0.05)”. Even a brief mention like “significant” with a figure reference is helpful, but tie it to the data. Also, for multi-group comparisons, clarify what is being compared: readers should know if “significant tumor regression” refers to compared to untreated controls, or compared to single-agent treatments. In Figure 2, for instance, explicitly noting that the triple therapy outperformed both single agents in text would be useful.

• “We observed with interest that, when tumors were treated with anti-CD47, there was an increase in total T-cell clonal expansion… and in the fraction of clones within their own repertoire as well as an increase in shared clonality between the tumor and SLN… This is consistent with an enhanced clonal T cell response generated in draining SLN with migration of T cells from the draining SLN to TME.” – This is an exciting finding about T-cell clonality, but the wording can be made more concise and objective. Remove “with interest” and simply state the observation: “We observed that anti-CD47 treatment increased overall T-cell clonal expansion and the fraction of dominant T-cell clones, as well as increased shared TCR clonality between tumor and SLN.” Then, ensure the interpretation is clearly linked: “This finding suggests a more robust, shared antigen-specific T-cell response developing in the lymph node and homing to the tumor.” Additionally, acknowledge sample size limitations here: it appears TCR-seq was done on one tumor/LN per group. It’s worth noting in the text or discussion that these clonality observations, while suggestive, are based on limited sampling (if true), to temper the interpretation.

• Figure references and labels: Double-check that every figure is cited in order and that lettering is consistent. For example, you mention “(Fig 3C)” and “(Fig 3D)” for MHC II and CD86 upregulation – ensure that those sub-panels indeed correspond to the described data (the text and figure legends seem to align well for Fig 3). Also, consider referencing supplemental figures in the Results text where relevant. You mention “S1 Fig 1C” in the legend for Fig 2 (line 1091–1092 in the PDF), implying there are supplementary figures. Make sure to cite them in the main text at appropriate points (e.g., if S1 Fig shows additional controls or data, say so in Results). This helps readers know supplemental data exists without having to find it on their own.

Discussion

• “Given these considerations, we designed a new combinatory therapy for our study.” – Minor wording: “combinatory” is not commonly used; “combinatorial” or “combined” would be clearer. For instance: “…we designed a new combinatorial therapy approach in our study.”

• “These data suggest a need for other types of therapies that are just as or more effective whether alone or in combination with ICIs [29].” – This sentence can be rephrased for clarity and grammar. Consider: “These data suggest that we need additional therapies that can be as effective as, or more effective than, current options – either used alone or in combination with ICIs [29].” This eliminates the colloquial “just as or more” phrasing.

• “Clinical trials have also demonstrated that sequencing of therapies are important, for example, when anti-PD1 is delivered before chemoradiotherapy (CRT), there is an abrogated immune response… whereas if anti-PD1 is delivered after CRT, there is better control… [29].” – There’s a subject-verb agreement issue (“sequencing … are important”) and the sentence is long. Split it for readability: “Clinical trials also demonstrate the importance of treatment sequencing. For example, delivering anti–PD-1 before chemoradiotherapy (CRT) can abrogate the immune response (due to radiosensitive CD8 T cells), whereas delivering anti–PD-1 after CRT yields better tumor control [29].” This way, the point about sequencing stands out, and the grammar is corrected (“sequencing… is important”).

• “CD47 has long been studied as a potential target… expressed on multiple human tumor types… allows tumors to evade the immune system by binding SIRPα, allowing tumor cells to express the ‘Don’t Eat Me’ signal to evade phagocytosis… [31].” – This background in the Discussion reiterates known information. While it is useful to a point, consider condensing it to avoid redundancy with the Introduction. For instance: “CD47 is an immune checkpoint that delivers a ‘Don’t Eat Me’ signal to phagocytes via SIRPα binding, and it is overexpressed in many tumors to enable immune evasion [19,31].” This single sentence could replace a few lines of text. Keeping the discussion focused on interpreting your findings in light of this mechanism (rather than re-explaining the mechanism at length) will make it more impactful.

• Interpretation of your results: The Discussion does a good job summarizing the experiments (CD47 KO, ALX301 + PD-1 therapy in two models, etc.) and tying them to the big picture. In a few places, the wording could be tightened. For example, “using both anti-PD1 and anti-CD47 showed significant response, and the addition of anti-CD47 enhanced survival” is a bit confusing because “using both” already implies anti-CD47 is there. You likely mean that in the PD-1–resistant model, anti-CD47 plus anti-PD1 yielded a significant anti-tumor response, and adding radiotherapy further improved survival. Make sure each combination’s benefit is clearly distinguished. You might rewrite that part as: “In our PD-1–resistant 4MOSC2 model, the combination of anti-CD47 with anti-PD1 produced a significant tumor response where anti-PD1 alone had minimal effect. Notably, adding tumor-directed radiotherapy to this combination further prolonged survival.” This explicitly states each component’s contribution.

• Tone down overgeneralization: While your results are promising, avoid overstating the translational impact. For instance, “the benefits of CD47 inhibition as a combination therapy reveal its promise as a future HNSCC treatment” is optimistic but could be softened to reflect that this is preclinical evidence. You might say: “…reveal its promise as part of future HNSCC treatment strategies, pending clinical validation.” Similarly, when concluding that you “provided a basis for rational combination of these therapies,” ensure you also acknowledge that further studies (especially clinical trials) are needed to confirm safety and efficacy in patients. PLOS ONE typically values conclusions that are supported by the data without speculation beyond what the results show.

• Expand on limitations: You appropriately mention some limitations (e.g., the short duration precluded toxicity assessment, and animal models don’t capture the full clinical complexity). It would strengthen the Discussion to elaborate briefly on these and any other limitations. For example, discuss the limitation of using only one tumor model for each scenario (4MOSC1 for responsive, 4MOSC2 for resistant – how generalizable might this be to other HNSCC tumors?). Also, the TCR sequencing was done on a limited number of samples; acknowledging that as a preliminary observation would be prudent. You could add: “Another limitation is the T-cell receptor sequencing analysis, which was performed on a single sample per group, providing a snapshot of clonality but not statistics. Future studies with larger sample sizes are warranted to confirm the observed clonal expansion patterns.” By openly discussing such issues, you align with PLOS ONE’s standards for transparency and encourage follow-up research.

• Language and style: Go through the Discussion for minor grammar issues and repetition. For instance, avoid starting sentences with conjunctions frequently (e.g., many sentences begin with “However,” – combine ideas if possible). Also ensure the past tense is used when referring to your results (“we found,” “we observed”) and present tense for general knowledge (“CD47 is…,” “these data suggest…”), which you generally do. One example: “CD47 inhibition has shown great results synergistically with other therapies as well.” Since this refers to past studies, it could be “CD47 inhibition has also synergized with other therapies in preclinical studies.” Small tweaks like this improve clarity.

Conclusions

Reviewer #7: 1. There are multiple typo errors that need to be edited by a native speaker.

2. The section "TCR Sequencing subsection" needs to be expanded.

3. Result should be re-written based on subtitles.

4. The comparison between ALX301 and ALX148 or other agents should be better described.

5. A tdRT-only arm and a CD47-only (without anti-PD-1) + tdRT group should be described or justified if omitted in the result section.

6. “anti-CD47,” “CD47 blockade,” and “ALX301” are used interchangeably.

7. Do statistical comparisons in TCR sequencing analysis.

8. Toxicity concerns of systemic CD47 inhibition in humans, and the immunological differences between mice and humans should be more discussed in the discussion.

**Do you want your identity to be public for this peer review?** For information about this choice, including consent withdrawal, please see our Privacy Policy

Reviewer #1: No

Reviewer #2: No

Reviewer #3: No

Reviewer #4: **Yes:** Zahra Sadin

Reviewer #5: No

Reviewer #6: No

Reviewer #7: No

---

## [Author Response · Author response to Decision Letter 1]

17 Jan 2026

Response to the Reviewers

Reviewer #2: Comment: Provide exact sample sizes (number of mice per group) for all experiments in the Methods section. Additionally, specify the statistical tests used for each figure panel (e.g., ANOVA/t-test) in figure legends or methods.

We thank the reviewer for this suggestion. The exact sample sizes for all experiments have been added to the corresponding figure legends, as some experiments had varying group sizes. Additionally, the specific statistical tests used for each experiment have been indicated in the figure legends.

Reviewer #3: General Comments:

This is an elegant and scientifically rigorous manuscript that explores the potential of CD47 immune checkpoint blockade in improving immunoradiotherapy responses in HPV-negative head and neck squamous cell carcinoma (HNSCC). The study is well designed, the models are appropriate, and the results are clearly presented.

Strengths:

Appropriate use of syngeneic murine models including both PD-1-sensitive and resistant tumors (4MOSC1 and 4MOSC2).

Integration of anti-CD47 therapy with PD-1 inhibition and tumor-directed radiation is timely and translationally relevant.

Immunophenotyping and TCR sequencing add strong mechanistic support.

Figures are informative and the manuscript is clearly written.

Optional Suggestions to Improve the Final Version:

The authors may consider adding a brief discussion of ongoing clinical trials involving CD47 inhibitors in HNSCC to enhance the translational context.

We thank the reviewer for this suggestion. We discuss current ongoing clinical trials and results that have been published. Page 17, Line 581: “It was found to have a favorable safety profile in clinical studies [20]. Additionally, in preclinical models, evorpacept combined with anti-PD1 led to favorable outcomes such as increased tumor growth inhibition and extended survivability [21]. While phase 2 clinical trials investigating evorpacept, pembrolizumab, and chemotherapy combination therapy for locally advanced HNSCC failed to meet primary endpoints, these findings suggest that additional or alternative therapeutic partners may be necessary for the synergistic potential of CD47 and PD1 inhibition (NCT05787639 and NCT04675333) [33].”

Moving the limitations paragraph earlier in the Discussion section might improve the overall flow of the manuscript.

We thank the reviewer for this suggestion. The discussion, including the limitations, has been edited to improve overall flow of the manuscript.

Overall, the manuscript is solid and presents impactful findings. I support publication without the need for revision.

Reviewer #4: Review Comments to the Author

Manuscript Title: CD47 Blockade Enhances Immunradiotherapy Response in Head and Neck Squamous Cell Carcinoma

Manuscript Number: PONE-D-25-34525

Dear Authors,

Your study on combining CD47 blockade with anti-PD1 therapy and radiotherapy in HPV-negative head and neck squamous cell carcinoma (HNSCC) is compelling and offers fresh insights into immunotherapy. Below, I’ve summarized my thoughts on the manuscript’s technical quality, data support, statistical rigor, data availability, and presentation, along with a few suggestions. I’ve kept this concise and focused, aiming for clarity and a natural tone.

1. Is the Manuscript Technically Sound?

Assessment: The study is solid, with a thoughtful design and reliable methods, though some missing details and text gaps need fixing.

You’ve done a great job laying out the goal—to test CD47 blockade (ALX301) with anti-PD1 and radiotherapy in 4MOSC1 and 4MOSC2 mouse models . These models mimic HNSCC well, and testing both responsive and resistant cell lines adds depth.

The CRISPR/Cas9 knockout , flow cytometry , and TCR sequencing are described clearly, with standard protocols that make replication possible. Verifications like Western blots are a nice touch.

The study follows UCSD IACUC guidelines, with clear animal welfare measures.

Text Gaps: Some sections cut off abruptly , making it hard to follow the full story.

We thank the reviewer for this helpful observation. We have added section titles and revised portions of the text to improve flow and ensure continuity throughout the manuscript.

2. Do the Data Support the Conclusions?

Assessment: Your data back the claims well, showing CD47 blockade boosts anti-tumor effects, especially with anti-PD1 and radiotherapy, though a few gaps need addressing.

The TCGA analysis linking high CD47 to worse survival sets a strong foundation.

Complete tumor regression in 4MOSC1 with CD47 knockout or combination therapy and better outcomes in 4MOSC2 with the triple regimen are convincing.

Increased MHC-II/CD86 expression and TCR clonality support enhanced immune activation.

You tie results to clinical needs, noting prior trial challenges.

Some immune markers (e.g., CD8 T-cell counts) weren’t significant (Page 17, Line 208), and TCR sequencing used only one sample per group (Page 29, Line 488).

We thank the reviewer for this observation. We have added statistical significance indicators to clarify which findings were and were not significant, improving transparency. Additionally, we have clarified in the text that the TCR sequencing results are based on limited sampling, providing supportive context to the flow cytometry findings.

Page 18, Line 633: “It is important to note that these observations, while suggestive of our hypothesis that ALX301 increases trafficking and migration of APCs, are based on limited sampling.”

Page 19, Line 681: “Another limitation is the T-cell receptor sequencing analysis, which was performed on a single sample per group. These results provide a snapshot of clonality to provide insight into the flow cytometry results of the tumors and lymph nodes, but not statistical results. Future studies with larger sample sizes are warranted to confirm the observed clonal expansion patterns with statistical comparisons.”

Report p-values for all key findings, including non-significant ones.

We thank the reviewer for this suggestion. All p-values, including non-significant ones, have been written out in each of the figure legends.

Justify or expand the TCR sequencing sample size.

We thank the reviewer for this suggestion. We have expanded and justified the low sampling size and acknowledged it as a limitation of this study.

Page 18, Line 633: “It is important to note that these observations, while suggestive of our hypothesis that ALX301 increases trafficking and migration of APCs, are based on limited sampling.”

Page 19, Line 681: “Another limitation is the T-cell receptor sequencing analysis, which was performed on a single sample per group. These results provide a snapshot of clonality to provide insight into the flow cytometry results of the tumors and lymph nodes, but not statistical results. Future studies with larger sample sizes are warranted to confirm the observed clonal expansion patterns with statistical comparisons.”

Clarify how results might translate to human patients.

We thank the reviewer for this suggestion. We have added to the discussion how our results might translate to human patients.

Page 20, Line 711: “Our results indicate that CD47 blockade, alone or in combination with anti-PD1 and radiotherapy, enhances dendritic cell activation and T-cell expansion in murine HPV-negative HNSCC models. The increased clonality and overlap between tumors and draining lymph nodes suggest that CD47 inhibition may influence antigen-specific immune trafficking between the tumor and lymph node, providing rational to lymphatic preservation. While these effects were observed in mice, they point to mechanisms that could inform the design of future clinical studies evaluating CD47-targeted combinations and treatment sequencing in HNSCC.”

3. Has the Statistical Analysis Been Performed Appropriately and Rigorously?

Assessment: The stats are solid for the most part, using the right tests, but clearer reporting on sample sizes and corrections would help.

T-tests, ANOVA, and log-rank tests fit well for tumor growth, immune data, and survival. Significant p-values support your claims.

You’re upfront about non-significant results, which builds trust.

GraphPad Prism 10 is a reliable choice.

Name the post-hoc test for ANOVA (e.g., Tukey’s).

Confirm data meet test assumptions or consider non-parametric options.

We thank the reviewer for this suggestion. The specific statistical tests used for each experiment have been indicated in the figure legends.

4. Have the Authors Made All Data Underlying the Findings Fully Available?

Assessment: You state all data are in the manuscript or Supporting Information (Page 7, Lines 7–8), which meets PLOS ONE’s rules, but I can’t fully verify without the files.

You say all data are available without restriction, aligning with journal policy.

In-Text Data: Tumor growth, survival, and immune results are summarized well , and TCGA data are public

Consider depositing TCR sequencing data in a repository like SRA.

We thank the reviewer for this suggestion. We are in the process of submitting the TCR sequencing data into SRA.

5. Is the Manuscript Presented in an Intelligible Fashion and Written in Standard English?

The manuscript is clear and well-written, with a professional tone, but typos and missing text need fixing.

We thank the reviewer for this observation. We have gone through the manuscript and have rewritten/reviewed the typos and missing text.

Reviewer #5: This manuscript presents important findings on the immunological enhancement provided by CD47 blockade in combination with anti-PD1 and radiotherapy in HNSCC models. It employs robust murine models (4MOSC1 and 4MOSC2), sophisticated CRISPR-based knockout systems, and insightful immune profiling. The translational implications are clear, especially for HPV-negative HNSCC, a subgroup with limited response to checkpoint inhibitors. However, the study would benefit significantly from additional validation in key areas, especially concerning TCR sequencing robustness, toxicity profiling, and therapeutic sequencing.

We thank the reviewer for their suggestion.

Regarding the TCR sequencing robustness, we have expanded and justified the low sampling size and acknowledged it as a limitation of this study. Page 18, Line 633: “It is important to note that these observations, while suggestive of our hypothesis that ALX301 increases trafficking and migration of APCs, are based on limited sampling.” Page 19, Line 681: “Another limitation is the T-cell receptor sequencing analysis, which was performed on a single sample per group. These results provide a snapshot of clonality to provide insight into the flow cytometry results of the tumors and lymph nodes, but not statistical results. Future studies with larger sample sizes are warranted to confirm the observed clonal expansion patterns with statistical comparisons.”

For toxicity profiling of anti-CD47, we acknowledged past literature that have explored the safety profile of CD47 inhibition and discussed toxicity profiling as next steps for future studies. Page 19, Line 671: “Past studies have demonstrated that CD47 blockade expresses minimal drug toxicity due to the lack of pro-phagocytic signals on healthy cells [34]. Throughout our study, no toxicity was observed in mice who received ALX301 therapy. It is important to note that while mice and humans share many aspects of cellular identity, key differences exist that impact the immune response to cancer and CD47 blockade [35]. Such differences include expression and binding of selectins, cytokine production, and Fc receptor expression [35]. Therefore, future studies that investigate the long-term toxicity profile of anti-CD47 are warranted to ensure safety during prolonged treatment in a clinical setting.”

For therapeutic sequencing, we have acknowledged prior studies that provide evidence on dosing and sequencing strategies that enhance therapeutic synergy. Page 9, Line 284: “A one-time dose of 4Gy was chosen for tdRT based on results that provided evidence that this dose of tdRT was not cytotoxic but still enhanced the percentage of CD8+ T cells and dendritic cell trafficking to the 4MOSC tumor [26]. With SBRT, there was an enrichment of tumor-specific T cells and CD8+ cytotoxic T cells, indicating an enhancement of an antitumor response [15]. Furthermore, we chose to deliver the tdRT before ICI administration due to evidence that delivering irradiation treatment directly to the gross tumor and sparing the cervical lymphatics can improve the systemic response that anti-PD1 and other immunotherapies provide [12,16].”

The manuscript shows increased MHC-II and CD86 expression but does not confirm whether DCs undergo full activation or cross-priming. Cytokine profiling or transcriptomic analysis of sorted DCs would strengthen the immunological conclusions.

We thank the reviewer for this insight. We have acknowledged in the text that while MHC-II and CD86 expression is a phenotype associated with DC activation, our studies do not confirm cross priming of tumor directed cytotoxic T cells by DCs.

Page 12, Line 402: “While increased MHC II and CD86 expression is consistent with a more activated dendritic cell phenotype, our study does not directly demonstrate cross-priming of tumor-specific cytotoxic T cells by DCs.”

Given the translational intent of the ALX301 combination regimen, the lack of toxicity profiling (e.g., hematological, hepatic, or general behavior outcomes) is a limitation. Basic safety data would bolster clinical relevance.

We thank the reviewer for this suggestion. We acknowledged past literature that have explored the safety profile of CD47 inhibition and discussed toxicity profiling as next steps for future studies. Page 19, Line 671: “Past studies have demonstrated that CD47 blockade expresses minimal drug toxicity due to the lack of pro-phagocytic signals on healthy cells [34]. Throughout our study, no toxicity was observed in mice who received ALX301 therapy. It is important to note that while mice and humans share many aspects of cellular identity, key differences exist that impact the immune response to cancer and CD47 blockade [35]. Such differences include expression and binding of selectins, cytokine production, and Fc receptor expression [35]. Therefore, future studies that investigate the long-term toxicity profile of anti-CD47 are warranted to ensure safety during prolonged treatment in a clinical setting.”

Only one tumor and SLN per group were sequenced, making the clonality findings anecdotal. Including replicates and statistical tests (e.g., diversity indices, clonotype frequency analysis) would solidify these claims.

We thank the reviewer for this suggestion. We have expanded and justified the low sampling size and acknowledged it as a limitation of this study. Page 18, Line 633: “It is important to note that these observations, while suggestive of our hypothesis that ALX301 increases trafficking and migration of APCs, are based on limited sampling.” Page 19, Line 681: “Another limitation is the T-cell receptor sequencing analysis, which was performed on a single sample per group. These results provide a snapshot of clonality to provide insight into the flow cytometry results of the tumors and lymph nodes, but not statistical results. Future studies with larger sample sizes are warranted to confirm the observed clonal expansion patterns with statistical comparisons.”

The manuscript lacks a clear rationale for using 4 Gy and applying it before ICIs. Comparative timing (pre vs post-ICI) would test hypotheses around immune priming and synergy.

We thank the reviewer for this suggestion. We have acknowledged prior studies that provide evidence on dosing and sequencing strategies that enhance therapeutic synergy. Page 9, Line 284: “A one-time dose of 4Gy was chosen for tdRT based on results that provided evidence that this dose of tdRT was not cytotoxic but still enhanced the percentage of CD8+ T cells and dendritic cell trafficking to the 4MOSC tumor [26]. With SBRT, there was an enrichment of tumor-specific T cells and CD8+ cytotoxic T cells, indicating an enhancement of an antitumor response [15]. Furthermore, we ch

---

## [Editor Report · Decision Letter 1]

22 Jan 2026

CD47 Blockade (ALX301) Enhances Immunoradiotherapy Response in HPV Negative Head and Neck Squamous Cell Carcinoma

PONE-D-25-34525R1

Dear Dr. Joseph A. Califano,

We’re pleased to inform you that your manuscript has been judged scientifically suitable for publication and will be formally accepted for publication once it meets all outstanding technical requirements.

Kind regards,

Amirreza Khalaji

Academic Editor

PLOS One
---

## [Editor Report · Acceptance letter]

PONE-D-25-34525R1

PLOS One

Dear Dr. Califano,

I'm pleased to inform you that your manuscript has been deemed suitable for publication in PLOS One. Congratulations! Your manuscript is now being handed over to our production team.

Kind regards,

on behalf of

Dr. Amirreza Khalaji

Academic Editor

PLOS One